

# Enhanced Quantitative Precipitation Estimation (QPE) through the opportunistic use of Ku TV-sat links via a Dual-Channel Procedure

Louise GELBART[1], Laurent BARTHES[2], François MERCIER-TIGRINE[1], Aymeric CHAZOTTES[2], and Cécile MALLET[2]

[1]HD Rain, 33 avenue du Maine, Paris, France
[2]LATMOS, UVSQ, Université Paris-Saclay, ´ Sorbonne Universite, CNRS, Guyancourt, France

**Correspondence:** Louise Gelbart (louise.gelbart@hd-rain.com)

**Abstract.** Earth – satellite microwave links such as TV-SAT can help for rainfall monitoring and could be a complement or an alternative to ground-based weather radars, rain gauges or satellites dedicated to Earth observation. Rain induced attenuation which is harmful for telecommunication is exploited here as an opportunistic way to estimate rain rate along the path link. This technique makes it possible to obtain rain measurements at a fine temporal resolution (a few tens of seconds) and with

a spatial resolution of few kilometers, which is a good compromise for human activities such as civil security (watershed monitoring, flash flood), agriculture or transport. Among the advantages of this technique, one can note the low cost of the hardware used (which can be commercial) as well as that of its maintenance on site. However, measured attenuation does not directly provide rain intensity and some parameters have to be estimated. Among these, it is necessary to take into account the contribution of the natural radiation of the atmosphere. In this paper, we detail a theoretical framework allowing to estimate

rainfall from the measurements of a low-cost sensor operating simultaneously over two parts of the Ku-frenquency band. Then this framework is assessed in a densely instrumented area in south of France, where it is shown that very good results are obtained when compared to rain gauges measurements, both in terms of overall rain accumulation and in terms of rainfall rate distribution. Then we apply this dual channel method in Ivory Coast, in the metropolitan area of Abidjan, where such an approach is very promising. It is shown that this technique when compared to rain gauge measurements give results far better

than a single-channel naive approach neglecting natural radiation of atmosphere, but that there are still significant errors in the rain assessment, leading to a persistent underestimation of rain accumulation. Finally we discuss various effects that could lead to this remaining underestimation, opening the door to further studies.

## 1 Introduction

The accurate measurement of small and medium-scale rainfall intensity is an important task, not only for understanding the

water cycle but, also for mitigating human and property damage or managing water resources. Traditional techniques for rainfall measurement include weather radars, rain gauges, disdrometers and remote sensing satellites. While satellites can be used to monitor precipitation on a global scale, they require a low Earth orbit to achieve a resolution of a few kilometers, resulting in low revisit time compared to rainfall dynamics. Ground-based weather radars cover smaller areas with higher revisit time,





but are costly to implement and maintain, while rain gauges offer point observations and require dense networks to properly

capture spatial variability, making deployment and maintenance complex and expensive, especially in difficult terrain.

Horizontal microwave telecommunication links such as cellular network stations have been widely studied for rainfall estimation (Messer et al., 2006; Goldshtein et al., 2009; Overeem and Uijlenhoet, 2016) notably in West Africa (Gosset et al., 2016; Turko et al., 2021) where such techniques are promising because few radars are available despite regular extreme rainfall and flooding events. However, they may face operational constraints (Chwala and Kunstmann, 2019; Polz et al., 2023) limiting

their use such as poor temporal resolution, coarse measurement levels, or difficulties in accessing data owned by telecom operators. In addition, their deployment can be limited, particularly in rural areas. Geostationary satellites, and TV-SAT satellites in particular, offer continuous downlink microwave sources covering most continents. TV-SAT broadcasts mainly use frequencies between 10.7 and 12.7 GHz (Ku-band), which are affected by absorption and scattering phenomena during rainy episodes (Barthès and Mallet, 2013; Colli et al., 2018). These phenomena lead to attenuation of the wave along the radio link, thus re-

ducing the power received by a ground-based station. Previous studies have shown the benefit of using such microwave satellite links for rainfall estimation. A full review of the various problems inherent in this technique can be found in Giannetti and Reggiannini (2021). The presence of rain on the path has a double effect: it attenuates the microwave signal from the satellite, as mentioned above, but it also increases the atmospheric emission through natural radiation from particles in the atmosphere (particularly raindrops) and hence the noise picked up by the receiving antenna. As a result, the received signal is a combination

of the signal from the satellite and the state of the atmosphere. In the presence of light rain and strong satellite signal, the signal received from the satellite is much greater than that of the natural atmospheric radiation. Under this condition, the latter can be neglected and rain-induced attenuation can be easily deduced by the difference between the signal received during rainy episodes and measured during dry periods occurring just before or after. In the presence of a higher rainfall rate, the signal from the satellite decreases substantially (10 dB or more) while atmospheric noise increases, and the previous assumption is no

longer valid, leading to an underestimation of rain-induced attenuation and, consequently, to an underestimation of the rainfall rate.

The present study aims to improve the estimation of precipitation made in various geographical contexts by low-cost equipment by taking into account the natural radiation of the atmosphere using a dual-channel measurement. This study follows that of Mercier-Tigrine et al. (2023), which introduced the sensor used for this study and the theoretical framework detailed in the

current paper. The objectives of this study are therefore to detail this theoretical framework and the hypotheses it presupposes, to validate it in a densely instrumented context, in France, and then to study its applicability in Ivory Coast in an operational context. We will also highlight the multiple sources of errors and inaccuracies inherent in the estimation of the rain intensity using Earth – satellite links.

Section 2 is dedicated to the physical context and defines the method used to estimate the transmissivity of rain by the means

of a dual-channel measurement. This section also presents the different sources of error to consider. In particular, we focus on a practical problem related to dual-channel measurement using a low cost (commercial LNB) device for which its characteristics can be significantly different depending on the channel used. Section 3 presents the dataset from two measurement campaigns





while Section 4 details the calibration procedures and the obtained results. Finally, we discuss in the last section the relative contribution of the different sources of errors and possible improvements.

## 2 Physical context

This section specifies the physical context and the inversion method proposed for precipitation estimation from ground satellite link. The Ku receiver used in this study (hereafter Ku device) is similar to a low-cost total power microwave radiometer (TPR) in which an antenna collects natural radiation emitted by atmospheric particle within a specific microwave band. This electromagnetic radiation is then amplified and filtered. The total power $P_{Tot}(t)$ measured at the device's output therefore

includes the power received from the TV satellite $P_{sat}(t)$, the power of natural radiation $P_{atm}(t)$ from atmospheric particles, including rain droplets radiations, and the power of the sensor noise $P_N$:

$$P_{Tot}(t) = P_{sat}(t) + P_{atm}(t) + P_N \tag{1}$$

The variations of the signal received from the geostationary satellites $P_{Sat}(t)$ are linked both to the signal emitted by the

satellite $P_E(t)$ and to atmospheric transmissivity $t_{atm}(t)$ while the variation of the natural radiations is linked to the antenna radiation temperature $T_A$. Hence the total power can be expressed as follow :

$$P_{Tot}(t) = P_E(t)G_E G_R G^i \frac{t_{atm}(t)}{l_{FSPL}} + T_A(t)kBG^i + T_N^i kBG^i \tag{2}$$

Where :

$P_E$ : Satellite transmitter power [W]

$G_E$ : Gain of satellite antenna

$G_R$ : Gain of Ku device antenna

$G^i$ : Gain of the low noise block-converter (LNB) of the receiver for a given frequency band noted i (for instance lower or upper Ku-band)

$l_{FSPL} = (\frac{4\pi d}{\lambda})^2$ : Free Space Path Loss ($d$ : distance between the satellite and receiver [m], $\lambda$ : wavelength [m])

$t_{atm}$ : Atmospheric transmissivity

$T_A$ : Antenna radiation temperature [K]

$k = 1.38.10^{-23}$ : Boltzmann constant [$J.K^{-1}$]

$k = 1.38.10^{-23}$ : Boltzmann constant [$J.K^{-1}$]

$B$ : Channel bandwidth of Ku device [Hz]

$T_N^i$ : Noise temperature of the receiver for a given frequency band noted i [K]





In the above equation, $l_{FSPL}$ and $B$ are constants and the gains $G_E$, $G_R$ and $G^i$ are assumed to be constant over time. Furthermore, $t_{atm}(t)$ and $T_A(t)$ are assumed to have identical values in various meteorological conditions regardless of the channel used provided that their central frequency is close to each other Barthes et al. (2003).In Eq. (2) the atmospheric transmittivity $t_{atm}$ can be expressed as:

$$t_{atm}(t) = t_0(t) t_R(t) \tag{3}$$

In which $t_R$ is the transmittivity induced by rain droplets ($t_R = 1$ in non-rainy situations) while $t_0$ is the transmittivity induced by other atmospheric components. The objective is to estimate the rain transmittivity $t_R$ to assess precipitation from the measurement of $P_{Tot}$.

Equations (2) and (3) show the impact of rain as highlighted by Giannetti and Reggiannini (2021). Rainfall has a twofold influence:

- it reduces $t_R$ and hence the total transmissivity $t_{atm}$ of the atmosphere leading to a decrease in the received signal $P_{sat}(t)$;

- simultaneously it increases the antenna radiation temperature $T_A$ resulting in an augmentation of the received signal $P_{atm}(t)$.

## 2.1 Atmospheric transmissivity $t_{atm}$

Various processes characterize atmospheric transparency, influencing the propagation of microwave satellite links. The total atmospheric absorption coefficient $k_{atm}(f, z, t)$ for frequency $f$ at height $z$ and time $t$ consists of contributions from atmospheric gases, clouds, and precipitation. From a zenith angle $\theta$ smaller than 70° (which is our case), a spherically stratified atmosphere can be approximated by a planar atmosphere. The optical depth $\tau$ of the atmosphere between ground altitude 0 and z is given by Mallet and Lavergnat (1992):

$$\tau(f, z, \theta, t) = \sec(\theta) \int_0^z k_{atm}(f, u, t)\, du \quad [\text{Np}] \tag{4}$$

And:

$$k_{atm}(f, z, t) = k_R(f, z, t) + k_0(f, z, t)$$





Where $k_R$ denotes the contribution of precipitation and $k_0$ the other contributions (gases and clouds). The atmospheric transmissivity for the entire atmosphere is defined by :

$$t_{atm}(f,\theta,t) = \exp(-\tau(f,\infty,\theta,t)) \tag{5}$$

In decibels, the total atmospheric loss factor denoted $A(f,\theta,t)$ is called atmospheric attenuation:

$$\begin{aligned} A(f,\theta,t) &= -10\log(t_{\text{atm}}(f,\theta,t)) \\ &= -10\log(t_R(t)) - 10\log(t_0(t)) \\ &= A_R(t) + A_0(t) \quad [\text{dB}] \end{aligned} \tag{6}$$

Where $A_R(t)$ is the rain attenuation and $A_0(t)$ is the other contributions.

## 2.2 Antenna radiation temperature $T_A$

In a non-scattering environment, a blackbody at a nonzero temperature $T$ radiates in the microwave region electromagnetic

energy at frequency $f$ given by the brightness intensity $B_{bbf} = 2kTf^2/c^2$ with $k$ the Boltzmann's constant and $c$ the velocity of light. This relation can be generalized to non-blackbody environment as : $I_f = 2kT_Bf^2/c^2$ in which $T_B$ defines the brightness temperature of the non-blackbody. By integrating over frequencies, these relations allow to deduce the power $P$ received in a bandwidth $B$ by an antenna in a perfectly absorbing and emitting chamber as $P = kBT_B$.

One can note that for a non-blackbody environment brightness intensity $I_f$ is lower than that of blackbody ($I_f < B_{bbf}$) and

thus $T_B$ is lower than the physical temperature $T$. Assuming a plane-parallel atmosphere, the antenna radiation temperature $T_A$ is given by Ulaby et al. (1981):

$$T_A(f,\theta) = \nu T_B(f,\theta) + (1-\nu)eT_G \tag{7}$$

Here $T_B$ is the brightness temperature of the atmosphere, $T_G$ the physical ground temperature, $e$ ground emissivity, and $\nu$ is the fraction of the antenna's radiation pattern directed towards the atmosphere.

Practically antenna and receiver are not lossless. Considering losses, we obtain :

$$T_A(f,\theta) = \nu'\nu T_B(f,\theta) + (1-\nu)eT_G + (1-\nu')T_0 \tag{8}$$

Where $T_0$ is the physical temperature of the receiving system, and $\nu'$ the radiation efficiency. For an ideal antenna and receiving system with ($\nu' = \nu = 1$), Eq. (8) reduces to $T_A(f,\theta) = T_B(f,\theta)$.

In a non-scattering atmosphere, the sky brightness temperature $T_B(f,\theta)$ measured at the ground surface is given by the radiative



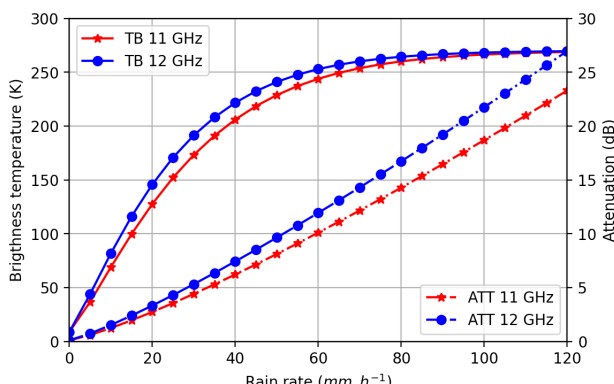

**Figure 1.** Brightness temperature of the sky and attenuation at 11 and 12 GHz and zenith angle of 45° as a function of rain rate.

transfer equation (Ulaby et al., 1981; Chandrasekhar, 2013):

$$
\begin{aligned}
T_B(f,\theta) \quad &= T_c t_{atm}(f,\theta) \\
&+ sec(\theta) \int_0^{+\infty} T(z) e^{-\tau(f,z,\theta)} k_{atm}(f,z)\, dz \quad \text{[K]}
\end{aligned}
\tag{9}
$$

Here $T(z)$ is the physical temperature at height z and Tc is the cosmic background temperature. It can also be written :

$$
T_B(f,\theta) = T_c t_{atm}(f,\theta) + T_m(1 - t_{atm}(f,\theta))
\tag{10}
$$

$T_m$ is the effective temperature and depends on the atmospheric profile and the frequency. In the literature, $T_m$ is generally
fixed around 280K but as shown in Fig. 1 in Barthes et al. (2003) the effective temperature depends on atmospheric profiles.
For the considered frequencies, neglecting the scattering effects in favor of absorption is only valid in a very limited range of
(low) rain rates. Scattering effects of rain droplets are considered in simulations (Barthes et al., 2003), where the brightness
temperature is studied in relation to atmospheric attenuation for different precipitation rates, showcasing the variability of natu-
ral radiation in the atmosphere. Figure 1 shows the variation of brightness temperature at 11 and 12 GHz and the corresponding
atmospheric attenuations for a homogeneous rain layer at different rain rates for a zenith angle of 45°. For rainfall rates ex-
ceeding approximately 40 mm/h, a saturation phenomenon is clearly observed in brightness temperatures while the attenuation
continues to increase. This occurs as the optical depth (Eq. (4)) of the lower layers increases, causing a significant attenuation
of radiation from the upper layers of the atmosphere.

## 2.3 Dual channel retrieval of rain transmissivity and the underlying hypotheses

In this study, the rain attenuation is derived using two channels, noted $A$ and $B$. These two channels are typically characterized
by the use of two different frequencies or polarizations. To quantify the impact of rain on the received signal, we use the ratio of
the differences of the received power between the two channels in rainy situations ($P_{Tot}^A$ and $P_{Tot}^B$) to the differences observed





$$\frac{P_{Tot}^A - P_{Tot}^B}{P_{Tot_0}^A - P_{Tot_0}^B} = \frac{G_E G_R \frac{t_0 t_R}{l_{FSPL}}(G^A P_E^A - G^B P_E^B) + (G^A - G^B)kBT_{A_R} + Bk(G^A T_N^A - G^B T_N^B)}{G_E G_R \frac{t_0}{l_{FSPL}}(G^A P_E^A - G^B P_E^B) + (G^A - G^B)kBT_{A_0} + Bk(G^A T_N^A - G^B T_N^B)} \tag{11}$$

in non-rainy situations (just before and/or after the rain event, $P_{Tot_0}^A$ and $P_{Tot_0}^B$). Our experience has shown that it is necessary to define two different gains $G^A$ and $G^B$ for each channel of the low cost LNB as they can be different from one channel to another. This is especially true if one channel is in the lower TV-sat band (10.7-11.7 GHz) and the other in the upper TV-sat band (11.7-12.7 GHz). These assumptions coupled with Eq. (2) lead to Eq. (11).

For an ideal LNB with $G^A = G^B$ and $T_N^A = T_N^B$ this equation reduces to $t_R$:

$$\frac{P_{Tot}^A - P_{Tot}^B}{P_{Tot_0}^A - P_{Tot_0}^B} = t_R \tag{12}$$

Equation (12) allows the estimation of rain transmissivity ($t_R$) and consequently the attenuation due to rain ($A_R$) through Eq. (3) and Eq. (6), given the knowledge of the reference level $P_{Tot_0}^i(t)$, also referred to as baseline signal or clear sky reference. Due to the high temporal variability of rain $P_{Tot}^i(t)$ exhibits distinct temporal characteristics depending on whether rain is present on the link. Thus, it becomes possible to identify rainy and non-rainy periods and hence to deduce $P_{Tot_0}^i(t)$. In this study a method similar to the method developed in Barthès and Mallet (2013), is used to separate rainy and non-rainy periods (see details in Sect. 3.2). As $P_{Tot_0}^i(t)$ cannot be observed during rainy conditions, interpolation is performed between values of $P_{Tot_0}^i(t)$ observed just before or just after the rain during non-rainy conditions (see Fig. A2 in the section about results for an illustrative case).

we list the different factors that affect the estimate of $t_R$, so we list the various effects :

1. Baseline Estimation Error:

   Rain/no rain detection errors can be caused by a number of factors, like changes in atmospheric composition (gases and clouds) during rain event. Estimating the baseline is more difficult in case of slight signal attenuation for example during long stratiform events.

2. Dual frequency

   Equation 12 show that it is necessary to use two channels with different characteristics. Ideally, one should receive a standard satellite level while the other one should be tuned to a channel where it receives mainly atmospheric radiation and no (or almost no) satellite signal. This approach helps mitigate any dependency on frequency differences and ensures accurate estimations. This implies as well that the $t_R$ of Eq. (11) corresponds to the one of the channel receiving strong signal.





3. Saturation:

The power received from natural radiation during heavy rain events reaches a limiting value (Fig. 1),known as saturation. This due to the fact that only the lower part of the atmosphere contributes to the signals received by both channels and therefore not representative of the complete state of the atmosphere. At the same time, satellite signals tend to fade because the attenuation is strong, leading to increased error in $t_R$ estimation.

4. Gain Channel:

In practice, due to the use of low cost LNB the gains between the two channels could be slightly different. In this case Eq. (12) should not be used directly (see Sect. 4.1). We introduce the calibration parameter $\alpha$ such as $\alpha_G = G^A/G^B$ also referred as $\Delta G = G_{dB}^A - G_{dB}^B$. For further details on $\alpha_G$, its estimation and the associated calibration procedure see Sect. 4.1.

## 2.4   Rain Retrieval

The estimation of rain rate from rain attenuation $A_R$ is based on ITU-R recommendations. The following relationship between lineic attenuation $\gamma_R$ and rain rate is given by ITU-R P.838-3 (2005) :

$$\gamma_R = kR^\alpha \quad \text{[dB/km]} \tag{13}$$

Where $k$ and $\alpha$ are two coefficients depending on frequency, polarization and elevation angle, and $R$ is the rain rate in $mm.h^{-1}$.

The total attenuation along the link of length $L$ crossing the rainy zone is therefore:

$$A_R = k \int_L R(l)^\alpha \, dl \quad \text{[dB]} \tag{14}$$

If the rain layer is assumed to be homogeneous vertically and horizontally then Eq. (14) reduces to:

$$A_R = kR^\alpha L \tag{15}$$

This relationship makes it possible to estimate the rain rate from the rain attenuation.

Several sources of uncertainty exist. In Eq. (13), the coefficients $k$ and $\alpha$ depend also on the microphysics of rain (i.e. raindrops size distribution), particularly below 9 GHz. In our case (11-12 GHz) we can expect a slight dependence. Another source of error concerns the use of Eq. (15) instead of Eq. (14). Indeed, Eq. (15) is based on the assumption of a homogeneous rain layer both horizontally and vertically. In practice, this assumption may not necessarily hold, and corrections need to be applied to the Eq. (15). Therefore, ITU-R P.618-12 (2017) introduces an equivalent rain cell through the use of a horizontal reduction factor

and a vertical adjustment factor in the calculation of $L$. The same concept is applied in Mello and Pontes (2012), where the authors introduce the idea of an effective rain rate determined empirically. Their conclusion points to a notable enhancement compared to the ITU approach. In Lu et al. (2018), the authors developed a model based on exponential rain cell profiles,





integrating a rain rate adjustment factor. This factor is then determined using the DBSG3 database. Their results reveal that this new model outperforms other existing models, including the ITU model, over various latitudes, frequencies and elevation angles.

The estimation of the effective rain height leads to another source of uncertainty. Indeed, effective rain height is more or less assimilated to the altitude $H_0$ of the 0°C isotherm. The ITU-R P.839-4 (2022) recommendation preconizes to increase $H_0$ by a few hundred meters (360m) to consider the melting layer, which is supposed to provoke a lineic attenuation larger than the liquid rain layer (Giannetti and Reggiannini, 2021). Because most of ITU models are based on annual database in the northern hemisphere, significant variability in $H_0$ and the melting layer height are not taken into account particularly in tropical regions. In Das and Maitra (2011), the authors propose to model $H_0$ in Indian region as a decreasing function of the rain rate while other studies have shown that "the effective rain height in the tropics could well be above the zero-degree isotherm liquid water" (see Green (2004) section 4.2 for more details). In the present study, the slant path length $L$ is simply estimated by:

$$L = \frac{H_0 - H_S + 0.360}{\sin(\theta)} \quad [\text{km}] \tag{16}$$

Where $H_0$ is the altitude of zero degree isotherm, $H_S$ the altitude of the ground sensor and $\theta$ the elevation angle of the ground - satellite link. It would be unreasonable to use a unique climatological freezing level height, especially in mid-latitudes where the mean freezing level ranges for instance in northern Italy from 1.5 km in January to 4 km in August (Giannetti et al., 2017). Using a seasonal average freezing level height would lead to large errors as well. For instance, the effective daily mean freezing level simulated by the ARPEGE model of Météo France (Bouyssel et al., 2022) in January 2023 in the area of Nice, south of France, ranges from 1.0km up to 3.1km, with 11 days above 2.5km and 9 days below 1.5km. This variability leads in this study to choose to use a dynamical freezing level height, determined from the vertical profiles of temperature simulated by the ARPEGE model and accessible in real-time from Météo France through an API.

Finally, we take into consideration the wet antenna effect which is a well-known issue in ground microwave links (Leijnse et al., 2008). Our conducted experiments have demonstrated an attenuation due to wet antenna ranges from 0.1 to 0.8 dB. So in this work, we will apply a fixed correction of 0.2 dB to mitigate this effect, while acknowledging the necessity for further research. This effect is particularly relevant for a light cold rain (minimal attenuation) but when attenuations are measured for tropical rainfall (typically greater than 10 dB) the phenomenon becomes negligible. In addition, this problem can be solved at least partially by placing protective cap over the horn.

## 3 Data Sets

### 3.1 Sensors and raw measurements

The sensors we use in this paper operate in the Ku frequency band, and are developed by the company HD Rain (hd-rain.com) for commercial purposes.





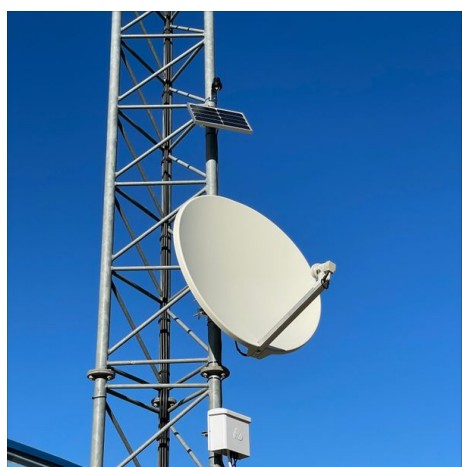

**Figure 2.** Image of a sensor installed by HD Rain on a mast of a fire brigade building (commercial collaboration with the SDIS, departmental fire and rescue service) with in the south of France

Figure 2 shows a Ku-sensor installed in the south of France. It is composed of a commercial dish able to retrieve and con-
centrate the signal on a low noise block downconverter receiver (LNB), connected through a coaxial cable to an electronic box measuring the power arriving from the satellite. The data are transmitted in real-time to a server using a SIM-card and a GSM antenna. The entire system is powered by a solar panel and Lithium-ion Battery, providing 10 days of operation on cloudy days. Each sensor sequentially measures the power (in dBm) in 4 different modes (4 channels) every 15 seconds: two polarizations: vertical and horizontal, and two frequency bands: lower frequency (10.7 - 11.7GHz) and upper frequency (11.7 - 12.75GHz).
The measurement is the total power received over this 1GHz-width frequency band. Some commercial TV-satellites emit no signal through an entire frequency band or polarization. In this scenario, the sensor behaves like a radiometer, capable of measuring atmospheric radiations.

**Table 1.** Table of designations used for HD Rain devices.

| Device | Ku-sensor | | | | | |
|---|---|---|---|---|---|---|
| Type | S | | SR | | | |
| Algorithm | Std | | Std | | Dual | |
| Polarization | H | V | H | V | H | V |

In this paper, these Ku-sensors are identified by a number: Ku sensor N°x. Additionally, sensors can operate in different
ways, summarized in Table 1. When the satellite targeted by the sensor emits signal over all channels, we use: Ku, S. where "Ku" signifies Ku sensor and "S" indicates that all four channels receive satellite signals. When two channels of one frequency band do not receive any signal, we use: Ku, SR. The "R" refers to the channels without a signal, behaving like radiometers. Moreover, two algorithms are used to retrieve rain rates from satellite signals (cf. Sect. 2). First, the dual-channel method consists in applying Eq. (12). As seen in Sect. 2.3, this approach can be used exclusively for SR Ku-sensors because it is necessary





to impose two channels having different characteristic, i.e one mainly measuring the power received from a TV-sat the other measuring atmospheric radiation. It will be referred to as "Dual". Secondly the standard method, currently employed for HD Rain's commercial products consists in applying Eq. (11) and neglect atmospheric radiation components to directly calculate $t_r$ as $P_{Tot}^A / P_{Tot_0}^A$. It will be called "Std". The two polarization are named "H" for horizontal polarization and "V" for vertical polarization. When not specified, horizontal polarization (H) is used. To name the rain gauges we use the following acronym :
"RG".

The results presented here come from such data from HD Rain Ku-sensors (validation data) installed in Abidjan, Ivory Coast and Cadarache, France, compared with nearby rain gauges (reference data).

### 3.2  Data treatment

The measurements made by the Ku-sensors consist of power levels of signals received at the satellite dish. These signals are composed either of a satellite signal and a radiometric component or solely of a radiometric component. In any case, estimating signal variations due to rainfall from such raw signals requires determining a baseline, i.e., a signal level that would be measured in the absence of rain (see Eq. (12), this baseline is the $P_{tot_0}$ component). This baseline estimation must be performed on both satellite ("S") and radiometric ("SR") signals. This baseline estimation challenge is common in the literature, for both mi-
crowave links (Schleiss and Berne, 2010) and satellite measurements (Barthès and Mallet, 2013; Gianoglio et al., 2023). In this work, we employ a machine learning approach. The algorithm used is similar in its objectives to the one presented in (Barthès and Mallet, 2013) but in an improved version: it is a Long Short Term Memory-type algorithm (Hochreiter and Schmidhuber, 1997) taking 6 hours of raw measurements as input and returning for each date a real number between 0 and 1 related to the probability that the given date is rainy. By applying a threshold to these values, we identify rainy periods, then assume that the
baseline is a segment connecting the last measurement before the start of a rainy period to the first measurement following that rainy period.

However, physically inconsistent situations can occur. During strong rainy events, transmittance tends toward 0 because $P_{Tot}^A$ and $P_{Tot}^B$ tend to be equal, so attenuation tends towards infinity. T To avoid physically impossible values or "NaN", we
define a threshold to force the transmittance not to exceed this threshold value : if $t_R < 0.005$, then $t_R = 0.005$. It may also happen that the transmittance exceeds the value 1, in which case we force the result to 1.

### 3.3  Installation in Cadarache, France

In collaboration with the French Alternative Energies and Atomic Energy Commission (CEA), seven Ku-sensors have been installed in a 30km zone in south of France, in a Mediterranean climate, at the end of 2022. The group of Ku-sensors is split
into two groups on Fig 3: three 'SR' sensors (marked in red) aimed at the Eutelsat 5W (E5W) satellite, which does not transmit a signal on the upper frequency band (11.7 - 12.7 GHz); four 'S' sensors (marked in green) aimed at the ASTRA 19 satellite,





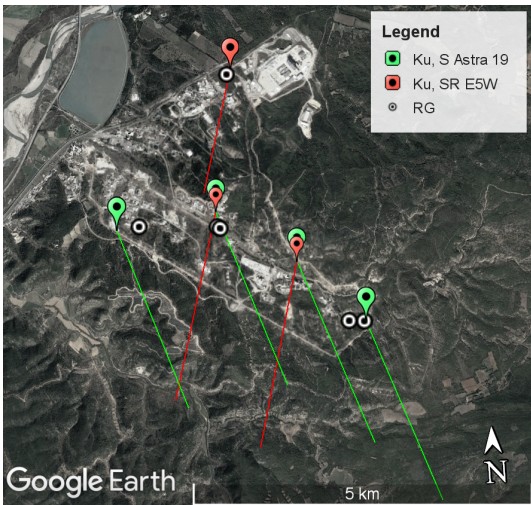

**Figure 3.** Map displaying devices available within the CEA area: Ku-sensors differentiated by the satellite they target, Eutelsat 5W (red marker) and Astra 19 (yellow marker). The satellite link path is identified with colored lines. Additionally, 1-hour resolution rain gauges are represented by points with an enclosed circle (From © Google Earth pro).

which emits signals on all channels. Moreover six rain gauges, each recording data with a resolution of one hour over a period of two and a half months, from April 2023 to June 2023 (white points), are installed in the same area.

### 3.4  Installations in Abidjan, Ivory Coast

The second area of interest is the metropolitan area of Abidjan, in Ivory Coast. About 156 Ku-sensors have been installed by HD Rain in this area (both in the city center and in the eastward area in order to measure rainfall before it reaches Abidjan) in 2021 to produce real time rain maps and nowcasts on the city. In this study, we will only focus on Ku-sensors installed close to a rain gauge. To do so, two groups of rain gauges are available. The first group consists of a 1-day resolution rain gauge located at Abidjan airport, close to three Ku-sensors (740 meters). These data cover a period ranging from 230 to 470 days. Two Ku-sensors target the satellite Eutelsat 36b, and the third Ku-sensor targets the satellite SES 5. The beams of these satellites transmit in the upper-frequency band of the Ku-band (11.75 - 12.75 GHz). Almost no signal is emitted over the lower band, rendering this channel as a radiometer. The second group of rain gauges consists of four 30-min resolution rain gauges over a period of 84 days(from May to July, rainy season), strategically positioned in four different districts of Abidjan. Five Ku-sensors targeting the satellite SES 5 and another six Ku-sensors targeting the satellite Eutelsat 36b (E36b) are located close to these rain gauges. Figure 4 displays all the instruments: Ku-sensors are distinguished by red and blue colors, corresponding to the targeted satellite, while rain gauges are represented in green (1-day resolution over an extended time period) or white (30-min resolution over a shorter time period).





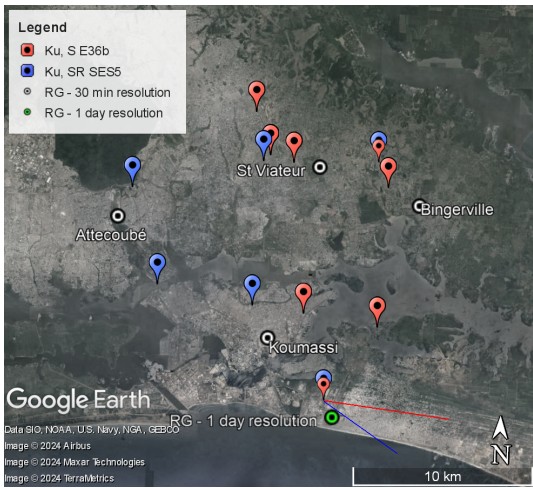

**Figure 4.** Map of available devices in the Abidjan area: Ku-sensors colored according to the satellite they target, E36b (red marker) and SES5 (blue marker); rain gauges represented by points with an enclosed circle: in white for the four 30-minute resolution rain gauges and in green for the Abidjan airport rain gauge at 1-day resolution (From © Google Earth Pro).

## 4  Results

This section is devoted to validating the two-channel algorithm for calculating precipitation by comparing the results of the
Ku sensors with reference data (rain gauges) and with the results of the algorithm using one channel (standard method). We first detail the calibration procedure used to estimate the LNB gain between two channels needed to apply the dual channel approach, then we show the interest of the approach over the densely instrumented site of Cadarache, before finally applying the method in Ivory Coast where the use of such an approach is critical for rain assessments.

### 4.1  Calibration procedure

As explained previously in Sect. 2.3, for the low cost LNB there is a gain offset $\Delta G$ between the two channels. When the Ku-sensor does not receive any signal from the satellite ($P_{Sat} = 0$) it is rather simple to estimate this parameter.

Under this condition, we can derive from Eq. (12):

$$\frac{P_{Atm}^A}{P_{Atm}^B} = \frac{G^A k B T_{A_R}}{G^B k B T_{A_R}} = \alpha_G \qquad (17)$$


Then $\Delta G$ can be easily evaluated by subtracting the two channels received powers (in decibel):

$$\Delta G = 10 log(\alpha_G) = P_{Atm,dBm}^A - P_{Atm,dBm}^B \qquad (18)$$





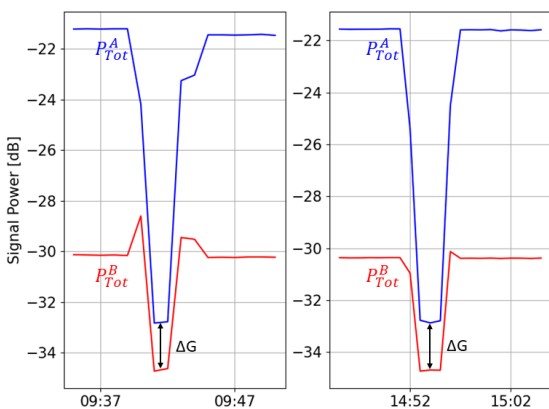

**Figure 5.** Raw signals measured by a Ku-sensor in France over two channels on the 20th December 2022 (left) and on the 20th February 2023 (right) allowing to assess the value of the gain offset.

Three procedures are developed for which the satellite signal is cancelled ($P_{Sat} = 0$). The first procedure involves shifting the antenna slightly so that it points away from the direction of the satellite. The second procedure consists of interposing an radio frequency absorber in front of the LNB while the third procedure consists of waiting for a sufficiently strong rain event when $t_R \approx 0$ (the satellite signal being strongly attenuated by the atmosphere). Note that the first two procedure require human intervention while this is not necessary for procedure 3. On the other hand, the latter requires a strong rain event and waiting a time before it occurs. In the next two subsections we illustrate how $\Delta G$ was estimated for our datasets, using procedures 1 and 3.

### 4.1.1 From direct measurements in Cadarache, France

At Cadarache, procedure 1 has been performed on the three available Ku-sensors. As explained previously we point the antenna towards clear sky, without any satellite signal. In this way, the two channels receive the same atmospheric emissions and the difference between the channels allows to estimate $\Delta G$ (in dB). This procedure has been carried out once for two Ku-sensors and twice for one Ku-sensor (N°749).

For this Ku-sensor N°749, the operation has been carried out twice in December 2022 and in February 2023 by aiming for clear sky for around five minutes. Figure 5 shows the signals associated with these two evaluations of $\Delta G$. Channel A (blue) corresponds (before the experiment) to the measurements over the lower Ku-band for satellite Eutelsat 5W, while channel B (red) corresponds to the upper band. While we expect the signal over both channels becomes equal when targeting clear sky, there is a remaining offset corresponding to the $\Delta G$. A numerical analysis of this experiment leads to these values for $\Delta G$: 1.85dB on the 20th December and 1.87dB on the 20th February. Even if measured at two different times we can see that these





two values hardly vary. We are thus confident that the difference between the two signals corresponds to the expected theoretical value of $\Delta G$.


To compare two of the three procedures described above, we also look for a saturation event over the area and found one occurring on October 30, 2023. By assuming in the same way that this should lead to record the same signals on both channels, modulo the value of $\Delta G$, we can estimate again $\Delta G$. Following this procedure 3, we get $\Delta G$ = 1.0 for Ku-sensor N°749. Similarly, we get for the other two sensors in the area:

- N°748 : $\Delta G_{p1}$ = 1.0dB and $\Delta G_{p3}$ = 0.7dB

- N°749 : $\Delta G_{p1}$ = 1.85dB and $\Delta G_{p3}$ = 1.0dB

- N°750 : $\Delta G_{p1}$ = 2.3dB and $\Delta G_{p3}$ = 1.6dB

Where $\Delta G_{p1}$ is the value obtained with procedure 1 and $\Delta G_{p3}$ with procedure 3.

The difference between both procedures varies from 0.9dB to 0.2dB. Equation (12) assumes that $T_N^A$ and $T_N^B$ are equal and that the radiation produced by the sensor is negligible. Except that this assumption is wrong, and after a few experiments we estimate the difference between $T_N^A$ and $T_N^B$ ($= \Delta T_N$) to be < 15K. Furthermore, it can be seen that when the brightness temperature $T_A$ is very low, as in procedure 1, the channel-dependent brightness temperatures of the noise are no longer negligible. Whereas in procedures 2 and 3, $T_A$ is high (>170 K) so the difference $\Delta T_N$ is negligible. We therefore use the
value from procedure 3.

### 4.1.2   From long-term data with saturation in Abidjan, Ivory Coast

For a remote sensor network already deployed on another continent it is not simple to apply either of the procedures 1 and 2. Since they have not been applied during the installation of the Abidjan Ku-sensors, procedure 3 will be used for these sensors. Like in the previous section and as it can be seen on Fig. 6, one channel (A, red) receives the signal emitted by the satellite
while the other channel (B, blue) receives almost no signal and so works in a radiometric mode. When it starts raining (around 20:00 on Fig. 6), the blue signal, composed only of atmospheric radiations, increases as $T_A$ increases, while the red signal, mainly composed of satellite emissions, decreases as $t_r$ decreases. Around 20:05, both signals reaches a plateau and almost no signal variation occurs during the next ten minutes. This event occurs during very heavy rainfall : $t_r \approx 0$ while $T_A$ reaches a plateau as it can be seen on Fig. 1. Then we can estimate $\Delta G$ as shown on the figure.
Over a prolonged period in Ivory Coast, there were several occurrences of saturation. In theory, by calculating the difference between the two signals, we can identify a plateau with a minimum value corresponding to Delta G. Nevertheless to avoid noise effects or punctual bugs in the received time series, and to ensure that the calculated $\Delta G$ is consistent and corresponds to saturation events, we define it as the 1st percentile of daily minimums of the signal difference over a period of at least 3 months.



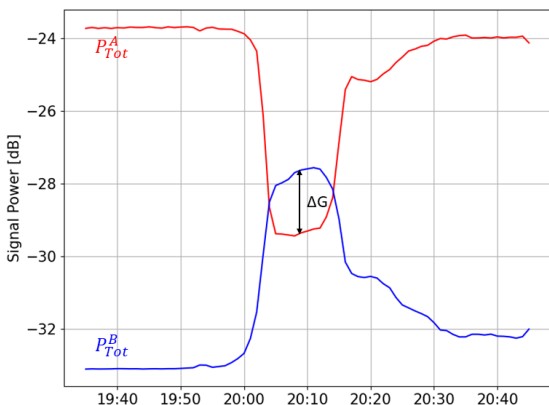

**Figure 6.** Raw signals from a Ku-sensor during a saturating rainy event: in red, the channel with satellite signal attenuated by the rain; in blue, the channel with very little signal and an increase in atmospheric noise due to the rain. The difference between these two signals in decibels corresponds to $\Delta G$.

Figure 7 illustrates this method. The upper graph shows the raw signals measured over two channels. The green curve represents the difference between both channels, considering only rainy periods. The value of $\Delta G$ calculated with 1st percentile approach described above is superimposed in black. As expected, we can see that it corresponds to a plateau (the calculated value is reached several times during the period). In addition, there is a dry season in August and September, which explains the need for sufficient data to calculate the $\Delta G$.

We then apply this procedure to 40 Ku-sensors in Ivory Coast, which leads to values of $\Delta G$ between +2dB and -3dB, centered around -1.5dB. Then it is important to assess the sensitivity of rain estimations to errors made on the estimation of $\Delta G$. To do so, we estimate cumulative rainfall for these 40 Ku-sensors (using Sect. 2.3) for the calculated $\Delta G$, as well as for this $\Delta G$ plus an offset varying in the range -0.9dB : +0.9dB. We choose this order of magnitude after testing the method and quantifying possible errors, especially by observing the difference we obtain in France between Procedure 1 and 3.

Figure 8 shows for each offset applied to the calculated $\Delta G$ the corresponding distributions of rainfall accumulations (in mm) for the 40 Ku-sensors. As expected, if $\Delta G$ is underestimated (negative offset), the gap between the signals received over both channels is overestimated and the rainfall is underestimated. On the other hand, if the $\Delta G$ is overestimated (positive offset), the two signals overlap more often, leading to unrealistic heavy rain intensities. Figure 8 shows the quite linear relationship between the increase of the $\Delta G$ and the increase of the cumulative rainfall. We nevertheless notice that the variations are significant but not critical. For instance, an offset of +0.5dB leads to a median rain overestimation around 10%, while an offset of +0.9dB leads to a median rain overestimation around 16%.





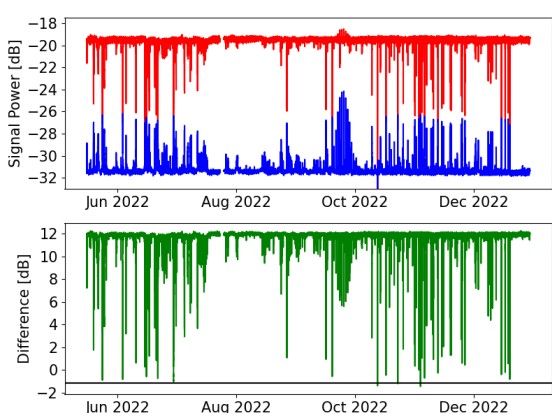

**Figure 7.** Upper graph: raw signals from Ku-sensor N°162. Red: $P_{Tot}^A$, meaning channel A including satellite emissions and corresponding to the upper part of the Ku-band in horizontal polarization. Blue: $P_{Tot}^B$, meaning channel B including only atmospheric radiations and corresponding to the lower part of the Ku-band in horizontal polarization. Lower graph: $\Delta G$ analysis. Green: difference between $P_{Tot}^A$ and $P_{Tot}^B$. Black: estimated $\Delta G$ (see details in the text).

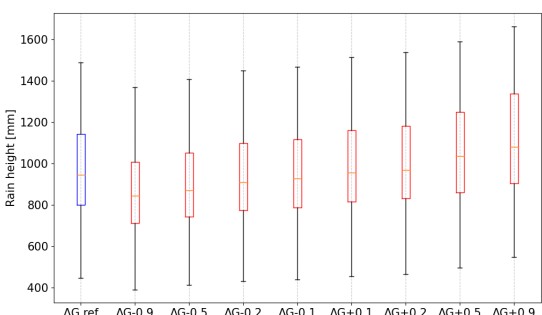

**Figure 8.** Distributions of cumulative rainfall obtained for 40 Ku-sensors in Ivory Coast over 8 months (May to December 2022). Blue: rain estimations made with calculated $\Delta G$. Red: rain estimations made when adding an offset to the calculated $\Delta G$.

For the rest of the study, $\Delta G$ is estimated by a dedicated analysis for each sensor during saturation events in order to min-imise the error on its estimation.


## 4.2 Statistical results in Cadarache, France (30-min rain gauges

In this section are presented the results obtained in Cadarache, France, from April to mid-June 2023, with the instruments detailed in Sect. 3 paring the different types of Ku-sensor (SR and S).



To begin with, we look at the rain accumulations for the Ku-sensors and the rain gauges. Figure 9b (right part) shows the
results when applying corrections to account for the effects of the melting layer and wet antenna, as detailed in Sect. 2.3 and
Sect. 2.4. Figure 9a provides, for reference, the results we would have obtained without considering these phenomena. The
measurements from the Ku-sensors appear similar to those of the rain gauges. In a scenario like this one, where the freezing
level is not very high and rain intensities are not always very strong, it emphasizes the necessity to account for both of the
mentioned phenomena. It is also noticeable that all rain accumulations are relatively dispersed (170 to 250mm for rain gauges,
180 to 260mm for Ku-sensors), considering the short distance between the instruments, indicative of highly heterogeneous
storms encountered in the south of France. Finally, it seems that the rain accumulations measured by the dual-channel Ku-
sensors (Ku, SR) are slightly better than those measured in standard mode (Ku, S), which could correspond to the fact that we
neglect the radiometric signal in the standard mode. We will revisit this point, which at this stage could also be attributed to
the natural variability of rainfall.

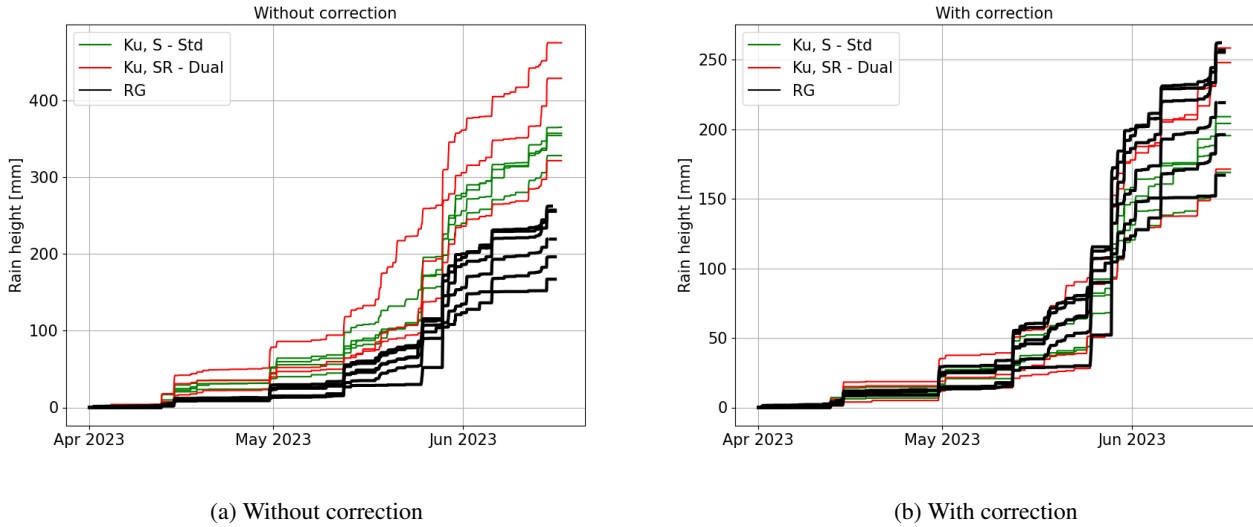

(a) Without correction                                        (b) With correction

**Figure 9.** Cumulative rainfall measured in Cadarache, France, from April to mid-June 2023 by various instruments and algorithms. Left: the
rain gauges (black), the SR Ku-sensors using the dual-channel method (red), and the S Ku-sensors using the standard method (green). Right:
The same, but with a correction applied to Ku-sensors measurements in order to take into account the impact of the melting layer and the
wet antenna (see Sect. 2.3)

Figure 10 shows the correlation between the different devices as a function of their distance and type (indicated by various
elements and colours). It can be seen that the correlation between the HDR stations is mainly above the trend line (84%),
indicating consistency between the HDR devices. In contrast, the correlation for the rain gauges, which provide a direct mea-
surement, is fairly heterogeneous, falling below 0.5 at a difference of 4 km. We also found a better correlation between rain
gauges and S-Std stations than with SR-Dual stations (58% of points below the trend line compared with 72%). But the cor-
relation between rain gauges and S-Std stations are more heterogeneous, as shown in Fig 10 the gap between the green points
(S-std - RG) and the trendline are on average wider than the red points (SR-Dual - RG). Overall, the devices follow the expected





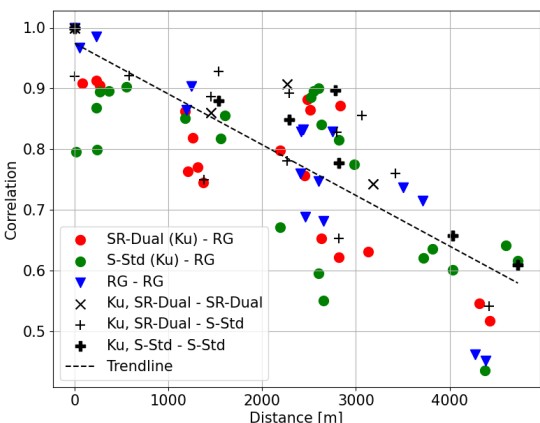

**Figure 10.** Correlations between all measurements made in Cadarache according to the distance between measuring instruments. See Table 1 for the legend and text for analysis.

trend and the correlation falls below 0.5 when the devices are separated by more than 4 km.

Figure 11 shows the quantiles of the 1h rain time series recorded by different Ku-sensors in comparison to the quantiles deduced from rain gauges measurements at the same resolution. Given the difference of nature between the rain gauges and the

Ku-sensors, obviously, we cannot expect similar values for both instruments. We suppose that these differences are largely mitigated by working at a 30-minutes resolution. On the contrary, it is clear that at high resolution, for instance 5-minutes, we would expect smoother records for the Ku-sensors.

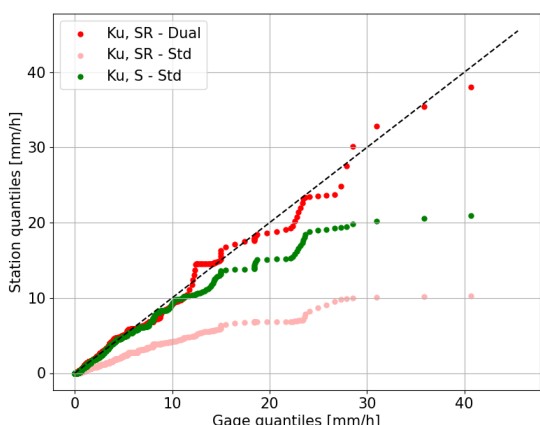

**Figure 11.** Quantiles (mm/h) from the 1st to the 100th percentile (coloured points) of 30 min resolution records for Ku-sensors vs. rain gauges, after excluding days when none of the devices detect rain-fall.



The first observation made from the Fig. 11 is the important difference resulting from the application of the dual channel method (red dots). The results of the Ku-sensors are drastically improved when compared with the one of the standard method (light red dots). When applying the standard method, we see that rain is strongly underestimated compared to rain gauges (light red curve far below the diagonal). For instance, the quantile corresponding to 20mm/h with rain gauges is measured around 7mm/h for the Ku-sensors. This is not surprising knowing that the satellite targeted by these sensors (Eutelsat 5W) emits a quite low signal, leading to a poorest signal to noise ratio (SNR). In this case, to not take into account the atmospheric radiation leads to strong underestimation, especially for strong rain. On the contrary, the quantiles of the SR-Dual Ku-sensors behave in a similar way to those of the rain gauges (red dots). We compute the linear regression line and find a directing coefficient of 0.96, which is very close to the trendline (ideal curve in black dotted lines). This proves the ability of the method developed in this paper to improve the results for such satellites where radiometry is directly accessible, and the very good results we obtain. Concerning the SR-Std Ku-sensors they show consistent results for rainfall <10 mm/h, but they tend to underestimate heavy rainfall. Again, this is not a surprise. For such sensors, we do not take into account of the atmospheric radiation (as for the SR-Std records), but we use a satellite that emits strong signal (generally around 16 to 20dB of signal-to-noise ratio in case of clear sky), so that the atmospheric radiations are largely negligible, except when the rain intensity becomes too high. Finally, we conclude from this figure that the dual channel method is capable of reliably reproducing the essential characteristics of the quantiles of rain gauge values and that this method improves the results of sensors with a radiometric channel and gives better results than S Ku-sensors.

It has to be noticed that the nature of the measurements made by both instruments is very different (punctual and sampled for rain gauge, integrated and continuous for Ku-sensor). Even if we work here at 30-min resolution to smooth these differences, we should be careful to not misinterpret any difference found in the final distributions. Nevertheless, for the S Ku-sensors, we can see that heavy rainfall (>15mm/h) are underestimated with Ku-sensors data, as seen before, due to the impact of the atmospheric radiations becoming significant. The atmospheric background noise is neglected while it significantly compensates the decrease of the signal when rain starts being heavy, leading to rain underestimation. For SR sensors for which atmospheric background noise is taken into account using the dual channel algorithm, results very close to those recorded by the rain gauges are obtained.

Finally, we conclude from this study that:

- The effects of wet antenna/melting layer appear to be adequately addressed with straightforward corrections. This would nevertheless need to be further investigated in future works, to assess it and even to improve its parameterization. This would also allow to ensure that other effects are not mitigated by this correction.

- The dual channel algorithm leads to very good results, consistent with rain gauges measurements, unbiased and quite equally distributed throughout the rain intensity range. When applied to satellite with low signal, this algorithm significantly improves the results.





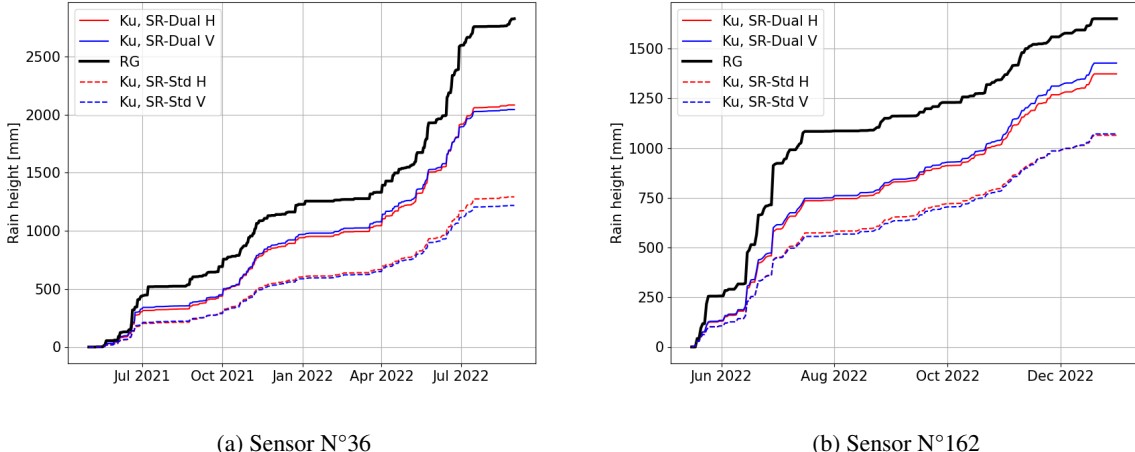

(a) Sensor N°36  (b) Sensor N°162

**Figure 12.** Rain accumulations using Ku-sensors with both methods (dual-channel and standard) and polarizations, compared to rain gauge measurements. (a) Ku-sensor N°36 compared to Abidjan airport rain gauge oveer 484 days. (b) Ku-sensor N°162 compared to the same rain gauge over 244 days. Rain gauge data are plotted with dashed black lines, Ku-sensors with dual-channel method are plotted with solid lines, and standard method with dashed lines; blue indicates horizontal polarization, and red indicates vertical polarization.

– The standard method (with atmospheric radiations not taken into account) applied to satellite with good SNR shows quite good results as well (no significant long-term bias visible on our dataset). However, it seems to underestimate heavy rainfall, which is logical since atmospheric radiations which are not negligible anymore are discarded in this case.

## 4.3 Results in Ivory Coast

Once our approach validated in a highly instrumented environment, we will apply it to a case study where it is much needed.
Indeed in the Ivory Coast signals received from the satellites are much weaker, so that an improvement of the standard method is needed.

### 4.3.1 Long term comparison (1-day rain gauges, Abidjan)

In this section, we compare two nearby Ku-sensors with the Abidjan airport rain gauge at a 1-day resolution. The goal is to observe the long-term behavior of the results using the dual-channel method applied to the sensors in a tropical zone.
On Fig. 12 the standard method (Ku, SR - std) is applied to both polarizations (red and blue dashed lines), and the dual-channel method is applied to both polarizations as well (solid red and blue lines) and compared with the Abidjan airport rain gauge (solid black line). Figure 12a shows the rainfall accumulations for Ku-sensor N°36 and the rain gauge over a year and a half (484 days), revealing a clear improvement in the results when using the dual-channel method. Specifically, the relative bias of Ku-sensors rainfall compared to the rain gauge decreases from 53% (Ku, SR - std H) to 24% (Ku, SR - dual H) for
horizontal polarization and from 55% (Ku, SR - std V) to 25% (Ku, SR - dual V) for vertical polarization. Similarly, Figure 12b




**Table 2.** Confusion matrices comparing all days with less than 50% NaN data on the two devices, the rain gauge and Ku-sensor N°162 over a period of 229 days and the rain gauge and Ku-sensor N°36 over a period of 488 days. A rainy day is a day on which the device detected precipitation; conversely, a non-rainy day is a day on which there was no rain. The matrix is normalize over the true conditions.

| Rainy day | N°162 | | N°36 | |
|---|---|---|---|---|
| RG vs Ku | Rain | No Rain | Rain | No Rain |
| Rain | 93.6% | 6.4% | 91.2% | 8.8% |
| No Rain | 20.6% | 79.4% | 28.9% | 71.1% |

represents the accumulations for Ku-sensor N°162 compared to the rain gauge over 8 months (244 days). The dual-channel method now exhibits a bias of 17% for horizontal polarization (Ku, SR - dual H) and 14% for vertical polarization (Ku, SR - dual V), indicating an improvement over the standard method biases of 36% for horizontal polarization and 35% for vertical polarization.

Using both channels (SR-dual on Fig. 12) strongly reduces the discrepancy with rain gauges measurements on total rainfall accumulations. A remaining underestimation is nevertheless still present. However, we did not detect any improvement in performance as a function of one polarisation or the other, we cannot state that one polarisation is better than the other and we assume that the impact is negligible in relation to the biases between the sensors and the rain gauges.

It is important to notice that both devices, the rain gauge and the Ku-sensor, do not observe the same phenomena. The Ku-
sensor indirectly measures rainfall integrated over a few kilometers long link, whereas the rain gauge directly measures rainfall at a given point on the ground. And this difference is all the more important in the case of mainly convective and heterogeneous events.

Table 2 displays the confusion matrix for sensors N°162 and N°36 in relation to the rain gauge. Each day is classified as Rain
or No Rain. First, the confusion matrix shows consistency in rainfall detection between both devices, with most days being classified in the same category by both instruments (N°162: 191 days out of 229 are well classified and accuracy is 85.2%; N°36: 399 days out of 488 are well classified and accuracy is 81.8%). Nevertheless, about 15% of the days show discrepancies between the instruments. For example, for station N°162, there are 32 days where the Ku-sensor detects rainfall while the rain gauge does not, as the rain may be too light to be detected by the rain gauge. And 7 days with rain gauge rainfall without
Ku-sensor rainfall.

Such a behavior was expected: Ku-sensors measure rainfall integrated over a link a few kilometers long while rain gauges give punctual measurements. If we suppose that both instruments are located at the same place, a shower can easily affect a portion of the link covered by a Ku-sensor without touching the rain gauge, while the reverse phenomenon is impossible (if the rain gauge is touched by a rain event, at least a part of the link will be touched as well). However, since this behaviour is
clearly not expected to be systematic (most rainfall events, especially the larger ones, should be large enough to affect both instruments) it should not affect the overall rainfall accumulation. Finally despite the improved results with the dual channel method, there is still an overall underestimation relatively to the rain gauges. The differences in measurements highlighted by the Table 2 are further discussed in the appendices.





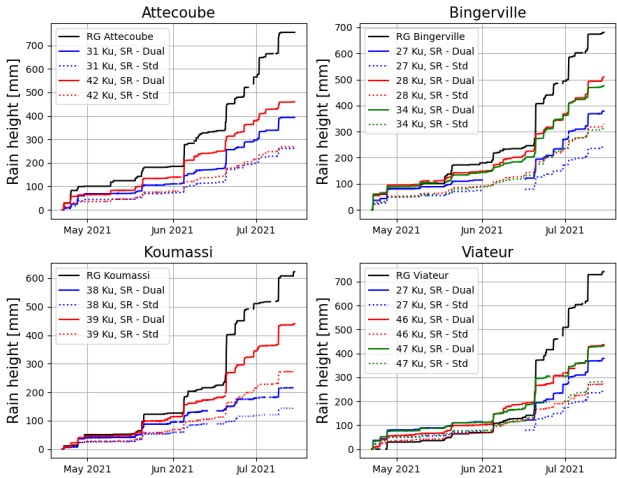

**Figure 13.** Cumulative rainfall recorded by rain gauges and Ku-sensors in Abidjan from late April to mid-July 2021. Each plot shows the records from a rain gauge and from all the Ku-sensors available at less than 4km from the rain gauge. For each Ku-sensor, both the standard and the dual-channel method are applied.

To conclude, the standard method applied to a weak satellite signal leads to a very significant underestimation of rainfall (of the order of 50%). However, by using the two frequencies of these satellites via the dual-channel method, the error is reduced but still present (15 to 25% in these cases). The impact of the different nature of the two instruments is analysed in the Appendix A. These results suggest that this difference in nature makes temporal comparisons between the measurements impossible, but does not explain the persistent difference observed between the instruments. The next section will then consist of a statistical study using 30-minute resolution data.

### 4.3.2 Statistical results (30-min gauges, Abidjan)

In this section are compared the measurements from four rain gauges and 12 Ku-sensors at a 30-minute resolution (see Fig. 4). Each rain gauge (Koumassi, St Viateur, Bingerville, and Attecoube) is compared with all Ku-sensors located in a 4km circle around the rain gauge. In all cases, 85 days of data are available, spanning from April to July 2021. We first observe the general behavior of the sensors on Fig. 13).

As in the previsous section we compare the cumulative data of four rain gauges to the values estimated from Ku-sensors. Similarly we note that the dual-channel method (solid colour line, Fig. 13) reduces the error compared to the standard method (dashed colour line, Fig. 13) but still falls short of the rain gauge results. Despite similar trends it is noticeable that the gaps between the curves widen during very intense events.



**Table 3.** Comparison of different rain sensors in Abidjan from late April to mid-July 2021. Each part of the Table (Attécoubé, Bingerville, etc.) corresponds to a rain gauge. The total rainfall indicated for each area (for instance 756mm in Attécoubé) corresponds to the total rainfall measured by this rain gauge. When compared to a Ku-sensor (each line of the Table), all dates when this sensor had missing values is excluded from the analysis, which leads to rain gauge accumulations different to the overall total (for instance 700mm for Attécoubé when compared to Ku-sensor N°42). For each Ku-sensor are indicated the percentage of missing values, the bias when compared to the rain gauge (negative means underestimation by the Ku-sensor), the root mean square error (RMSE), as well as the distance from the Ku-sensor to the rain gauge, and the satellite it targets. We notice that targetin SES5 implies a 5-km link, while targting E36b implies a 7-km link.

| Sensor | RG Rainfall.(mm) | Rainfall.(mm) | NaN (%) | Bias (%) | RMSE (mm/h) | Distance (km) | Satellite |
|---|---|---|---|---|---|---|---|
| Attécoubé - total rainfall = 756 mm | | | | | | | |
| No.42 | 700 | 450 | 7.4% | -36% | 2.05 | 1.70 | SES 5 |
| No.31 | 740 | 387 | 6% | -48% | 2.67 | 3.62 | SES 5 |
| Bingerville - total rainfall = 688 mm | | | | | | | |
| No.34 | 653 | 474 | 9% | -27% | 1.84 | 1.31 | ET36b |
| No.28 | 664 | 505 | 2.5% | -24% | 1.93 | 2.33 | ET36b |
| No.27 | 595 | 378 | 23.5% | -36% | 2.25 | 3.35 | SES 5 |
| Koumassi - total rainfall = 629 mm | | | | | | | |
| No.39 | 586 | 433 | 9.5% | -26% | 1.60 | 2 | SES 5 |
| No.38 | 428 | 214 | 36% | -50% | 2.40 | 4.87 | ET36b |
| St Viateur - total rainfall = 759 mm | | | | | | | |
| No.46 | 685 | 384 | 16.4% | -44% | 2.66 | 3.05 | ET36b |
| No.47 | 635 | 376 | 18% | -41% | 2.23 | 3.21 | SES 5 |
| No.27 | 656 | 335 | 27.5% | -49% | 2.85 | 3.23 | SES 5 |

In Table 3, we gather recorded accumulations by both of the rain gauges with values estimated through several Ku-sensors. It allows to assess quantitatively the results provided by the dual channel method. We note the influence of distance on the RMSE: for instance for Bingerville sensors, RMSE is increasing from 1.84mm/h for the Ku-sensor N°34 located at 1.3km of

the rain gauge up to 2.25mm/h for Ku-sensor N°35 located at 3.4km.

Distance seems to increase the error as well, which is less expected: Sensor N°42 exhibits a relative bias of -36% at a distance of 1.7 km, while Sensor N°31 shows a relative bias of -48% at a distance of 3.62 km. It is important to notice that the biases recorded here with dual channel approach seem larger than the ones recorded in the previous section with long-term daily measurements. This could be due to the fact that here we concentrate over the rainy season in Abidjan (May to July) when

most rain occurs as strong convective events with high rain intensities. As seen in France, these kind of events are more likely to lead to errors and saturation in Ku-sensors measurements, and so to stronger underestimation. The biases recorded here (25 to 50%) are then not incompatible with the long-term biases recorded at the airport (15 to 25%).

Figure 14 shows the correlation between the instruments according to the distance between them. The Ku-sensors are split ac-

cording to the satellite they target to emphasize the importance of the length of the satellite link. A consistent trend is recorded, showing a decrease in correlation as the devices move further apart. The correlation between the different Ku-sensors, regardless of which satellite they are pointed at, is good (red crosses, Figure 14). The trend in the correlation between the SES5 Ku-sensors and the rain gauges is the same as the global one (cyan dots, Fig. 14), with 55% of the points below the trend





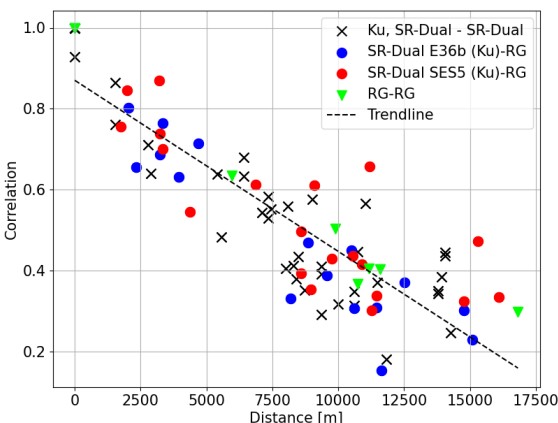

**Figure 14.** Correlations between instruments records according to their distance in Abidjan. Black crosses: Ku-sensors among themselves ; Green triangles: rain gauges among themselves; Blue dots: rain gauges vs. Ku-sensors targeting E36b; Red dots: rain gauges vs. Ku-sensors targeting SES5. The trend curve for all instruments using linear regression is plotted as a black dotted line.

line. In contrast, correlations between E36b Ku-sensors and rain gauges show 63% of points below the trend line. In other
words, correlations between Ku-sensors targeting E36b seem weaker than those of Ku-sensors targeting SES5, which could be
explained by the fact that E36B is lower in the sky than SES5 leading to longer path links (more than 7km).

Figure 15 shows the quantiles of the 30-min rain time series recorded by several Ku-sensors relatively to the quantiles re-
sulting from rain gauges measurements at the same resolution. For the standard and the dual channel methods, each of the
two curves correspond to an aggregate response of Ku-sensors targeting the same satellite. Figure 16 shows the ratio of rain
gauge quantiles to the quantiles from the Ku-sensors for both the standard and the dual channel methods, allowing to see the
differences between both instruments at different rainfall scales.

An enhancement in results is clear with the dual-channel method compared to the standard method (Fig. 15), particularly for
precipitation exceeding 20mm/h and even more prominently for rainfall surpassing 60mm/h. These results are similar to those
obtained in France (see Sect. 4.2): taking into account radiometric effects in the dual channel method strongly improves the
results.

In addition, we can see from Fig. 15 that up to 10mm/h there is very good agreement between the rain gauges and the ku-
sensors targeting SES5. Above this intensity, the error increases, but we note that it increases less on the sensors targeting SES5
than on those targeting E36b. This can be explained by a shorter link under the rain and therefore probably more homogeneous.

Finally, the results obtained from the dual channel method (SR, Dual) shows a general underestimation when compared to
the rain gauges, which confirms the underestimation already recorded on long-term rain accumulations (Fig. 15 and Fig. 16).
Because the instruments are different, we expect not to strictly follow the diagonal line, because of the spatial integration,

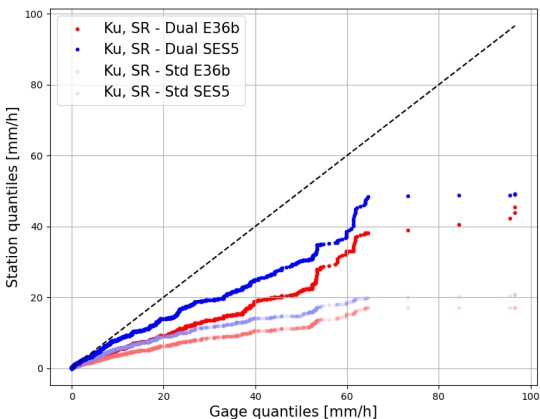

**Figure 15.** Quantiles (mm/h) from the 1st to the 100th percentile (coloured points) of 30-min resolution records for Ku-sensors vs. rain gauges in Abidjan, for both the standard and dual channel methods. All points are colored according to the satellite targeted by the Ku-sensors.

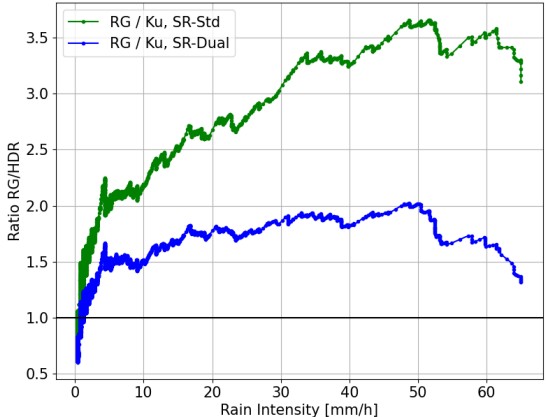

**Figure 16.** Ratio between the intensity quantiles of the rain gauges and the intensity quantiles of the stations using the dual channel method (blue) And the ratio between the intensity quantiles of the rain gauges and the intensity quantiles of the stations using the standard method in green, expressed as a function of the rain gauge intensities. A point above the 1.0 line at a given rain intensity means that for the gauge quantile corresponding to this intensity, the same quantile for the Ku-sensor is smaller.

Ku-sensors measurements are expected to be smoother than rain gauges measurements, and to underestimate high quantiles. Nevertheless, knowing that the data have been integrated at a 30-min time resolution, these differences are expected to be small, as they are in France (see Sect. 4.2). Figure 16 shows that the lowest quantiles are overestimated by Ku-sensors with the dual channel method (until about 1mm/h), which is probably normal given their difference of nature. However, this pattern swiftly shifts as intensities increase. We can see that the standard method leads to quantiles between 2 and 5 times weaker than





the values recorded on the rain gauges as the rain intensity increases, indicating an underestimation that increases with rainfall intensity. The dual channel method, on the other hand, shows a much more constant underestimation, never exceeding a factor of 2. This confirms the remaining underestimation of the dual channel method, but also that the intensity-dependence of this underestimation seems to have been solved. There is nevertheless a specific case that still needs to be treated, the saturation effect: we can see from Fig. 15 that the dual channel method cannot record rain intensities larger than about 40mm/h (E36B) or 50mm/h (SES5). This is due to the fact that there is no more signal available in this case. Nevertheless this concerns only a few extreme rain events and is probably not the cause of the overall underestimation, which concerns the entire rainfall intensity spectrum.

## 5 Conclusions

In this paper we proposed a dual channel method taking into account the radiometric component of the signal received by low cost Ku-sensors measuring the power received over the Ku-band from geosynchronous satellites emitting over an incomplete part of the Ku spectrum. The method makes it possible to obtain a good estimate of the rain rate even when receiving a weak signal from the satellite.

After studying the theoretical implications of such an approach, this method has been validated in a highly instrumented environment. In Cadarache, south of France, three Ku-sensors have been set to measure the signals received from a low-powered satellite. Four other Ku-sensors targeting high-powered satellites were already installed in the same area, together with five rain gauges. It has been shown that applying a standard algorithm (without taking into account the radiometric component of the signal) on the low-power satellites led to strong underestimation of rainfall, while using the dual channel approach allowed to retrieve very well the rain when compared to rain gauges. When compared to Ku-sensors targeting high-power satellites with standard approach, it has been shown that while the differences were very small for low rain rates, dual channel method outperform for high rain rates, when the radiometric component start to be non negligible even for high-power satellites.

Then, the dual channel method has been applied in Ivory Coast, where such a method is critical to provide low-cost, decentralized, robust rain measurements and where high power satellites are not available. Despite recurring errors leading to an underestimation of rain accumulations when compared with rain gauge measurements, we have considerably improved the results by taking into account the atmospheric background noise. We have shown the impact of the targeted satellite on the results, with better results when using a quasi vertically pointing satellite. We have shown that a standard method lead to strong underestimation very sensitive to rain intensity (with underestimation increasing as the intensity increases). When using the dual channel approach, the underestimation is weaker, and not clearly dependant on rain intensity.

However, and regardless of the main subject of the article, or at least in addition to it, two points remain to be clarified.





In France, it has been demonstrated that accounting for the effects due to wet antenna and the melting layer cannot be overlooked. Although errors related to precipitation estimation seem to be explained in order of magnitude by what is known about these phenomenon, further studies are needed to parameterize these phenomenon more precisely, which are relatively under-explored in satellite measurements.

In Ivory Coast, the approach presented in this paper significantly reduced the underestimation of rainfall, but did not completely explain it, far from it. After correction, underestimations of the order of 15 to 45% in cumulative rainfall are observed, seemingly evenly distributed across rainfall intensities. Therefore, it is necessary to investigate the causes that could explain this persistent underestimation.

As a conclusion to this paper, we will in the following paragraphs propose a set of possible causes of this underestimation. This will be done by referring to studies already conducted on the subject, assessing the likelihood that it is a plausible explanation, and/or suggesting experimental or numerical studies to test these causes.

**Non-linearity of k-R relationship** To convert attenuations into rainfall rates, we assume that rain is distributed homogeneously over the Earth-satellite link. Without considering for now the vertical structure of rain, this is naturally not true over the horizontal plane for links of a few kilometers length (a part of the link may experience heavy rainfall, while another part remains dry), causing an error as the attenuation-to-rainfall rate relationship is not linear (but power-law with a power coefficient slightly greater than 1 at 12GHz). However, it can be shown, on the one hand, that the error should lead to an overestimation of rainfall, and on the other hand, for a relatively extreme case (rainfall changing from 0 to 100 mm/h and then back to 0 over a 7 km link), than an overestimation of around 9% is expected. On average, in the long term, we therefore expect a significantly lower effect. This point, however, deserves consideration, which can be done, especially statistically, through the use of a rainfall simulator like Féral et al. (2003).

**Error on freezing height** The conversion of measured total attenuation to rainfall rate is directly linked to the altitude of the 0°C isotherm (freezing height). A bias of 10% in this height should lead to a bias of the same order in rainfall retrievals. This is independent of other associated errors (vertical inhomogeneity of rainfall, melting layer effects). The height of the freezing level used in this study are those predicted by the ARPEGE NWP model of Météo France. Although it seems unlikely that this effect alone explains the observed underestimation, especially in a region like the tropics where the height of the 0°C isotherm is relatively stable over time, it is worth testing this parameter. This can be done by comparing these estimates with measurements from vertically pointing radars, such as BASTA (Delanoë et al., 2016), or ROXI (Lemaître et al., 2016).

**Error about vertical structure of rain and associated effects**

In this study, rain is supposed to be vertically homogeneous from the ground up to the freezing height. Many phenomenon that occur over the atmosphere column can affect this hypothesis.





First the upper cloud limit can be under the freezing level, leading to a path length lower than expected, and so to real rain rates larger than calculated. Examples can be found in the literature about such events (Feng et al., 2014). It is unlikely that such events are sufficiently systematic to produce strong biases in tropics (strong convective events being generally associated to high-altitude clouds), but the occurrence of such phenomena can be explored more deeply.

Another phenomena, probably more frequent and affecting mostly convective rain is the fact that the freezing level is not always a clear limit (see for instance Giannetti et al. (2017) notably their Figure 6 and associated text): there can be liquid rain above the freezing height and solid ice below due to strong vertical winds. This could lead to systematic errors, especially in areas subject to heavy rainfall like the tropics. Another noticeable phenomena is the variation of the vertical (gravitational) velocity of rain drops with height: droplets fall faster at high altitudes, where the air density is low, and slow down as they
approach to the ground (Foote and Du Toit, 1969; Atlas et al., 1973). If we suppose that the rain rate is conserved through the column, this leads to a greater droplet density close to the ground, and so to a larger specific attenuation close to the ground when compared to higher altitudes. The average attenuation over the column is then lower than the real attenuation at the ground level, leading to an underestimation of the rain rate. This phenomenon is more important in case of high freezing level like in the tropics, and should be taken into consideration in future works. The last potential phenomena of interest is
the variation of the rain rate among the column under the effect of evaporation or condensation, as well as a variation of the droplet size distribution (under these effects or other ones like coalescence or breakup, see for instance Mercier et al. (2016)). This phenomenon could explain (a part of) the underestimation we notice only in case of systematic condensation (leading to a ground rain rate higher than the average one), occurring only in Ivory Coast, which is unlikely.

**Error on k-R relationship** In this work, we use an ITU attenuation - rainfall relationship. Many other parameterizations have been assessed, some of them dedicated to tropical systems (Moumouni et al., 2018).
Even if it is hard to robustly compare the different relationships that can be found in the literature, it would be useful to quantify the uncertainty associated with this phenomenon by applying several parameterizations and estimate the variability it introduces in resulting rainfall.

    **Radiometric saturation** The last and already mentioned phenomena that leads to underestimation is the signal saturation, which occurs when the satellite signal starts being negligible against the atmospheric radiations, so that the dual channel approach leads to rain transmittance tending to 0, so that it is no more usable. A systematic assessment of the underestimation due to this phenomenon, which occur more often in Ivory Coast, is needed even if it is not able to explain the underestimation
noticed for all rain intensities.

*Code and data availability.* For more information on the codes and the data, contact Louise Gelbart at louise.gelbart@hd-rain.com





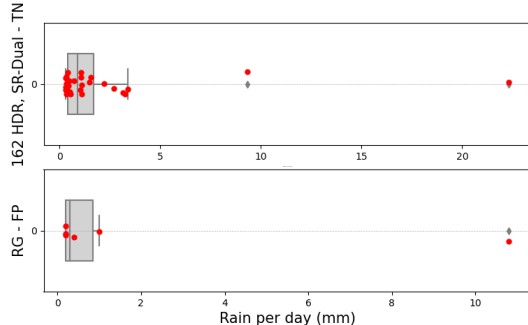

**Figure A1.** Top: distribution of Ku-sensor N°162 daily rainfall for the days when the Ku-sensors detects rain while the rain gauge does not (using the dual-channel method and horizontal polarization). The 32 rainy days are marked in red, and their distribution is represented by the gray box plot. Bottom: same for the rain gauge rainfall for the days when KU-sensor N°162 does not record any rain.

## Appendix A:  Long term comparison - difference between devices

We saw in the Sect. 4.3.1 that there are differences between the instruments, and here we seek to detail and understand these differences. Knowing that in this experiment both instruments are not located exactly at the same place makes the analysis more complex, but should not lead to any long-term differences, as long as there is no climatological variations of rainfall over the area of interest (which is not likely over a few kilometers in this plain area), and as long as the overall long-term rain accumulation is not critically determined by a few strong event (that could affect one instrument and not the other one).

So it is important to closely analyze these discrepancies. The aims of this analysis are:

- to see if these discrepancies lead to important errors.

- for days with significant discrepancies, to analyse whether the discrepancies stem from very localised events affecting one instrument but not the other, or from errors in the analysis algorithms.

More precisely, we examine on Fig. A1 the distribution of Ku-sensor daily rainfall for the days when the Ku-sensors detects rain while the rain gauge does not. Most differences correspond to daily rain accumulations from 0.25mm to 5mm (low rainfall): the median is 0.55mm. However, we observe two days with rainfall above 5mm, respectively 9.34mm and 22.32mm. Similarly, among the 6 days when only the rain gauge detects rain, only two large values (>10mm) are noticeable. This shows that the long-term bias between the two instruments, which for Ku-sensor N°162 corresponds to a rain accumulation difference of around 300mm, cannot be explained solely by specific events measured by one of the sensors and not by the other.

Finally, we show on Fig. A2 the raw signals from Ku-sensor N°162 corresponding to the strong rain events measured by this sensor but not by the rain gauge seen on Fig. A1. It is clear on the raw measurements that there is rain occurring over the link for both cases, which confirms that most of the discrepancies between both instruments are likely to be due to physical differences (rain heterogeneity) and not in algorithm issues. This will convince us in the rest of the analysis to not directly compare time-to-time measurements (with indicators like RMSE) but to statistically compare the results, in terms of overall





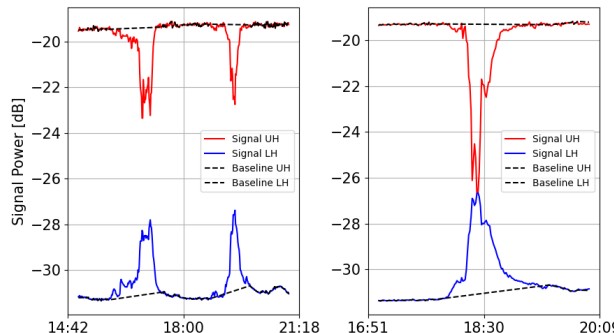

**Figure A2.** Raw signal measured by Ku-sensor N°162 for the two days when this sensor measured more than 5mm of rain while the rain gauge recorded no precipitation. Left: November 2, 2022. Total measured rainfall is 9mm. Right: November 18, 2022. Total measured rainfall is 22mm. Red: raw signal of the upper frequency band in H polarization. Blue: raw signal of the lower frequency band in H polarization. Dotted black: estimated baseline.

rain accumulation and rain intensity distributions.


*Author contributions.* LB, FM, AC, CM and LG design the study. LG and FM carried out the study with the contribution of LB, AC, and CM. LG prepared the article with contributions from all co-authors. More precisely AC CM and LB have written the introduction and the Physical Context; LG have written all the Results and the Data Sets section; and FM have written the Conclusions. All co-authors revised and approved the lastest version.

*Competing interests.* The authors François Mercier-Tigrine and Louise Gelbart are employees of HD Rain, a company that develop and build the sensor used in this article and sell products notably based on its measurements.

*Acknowledgements.* First of all, we would like to thank the CEA at Cadarache, and Dr. Thierry Hedde in particular, for funding the installations of HD Rain stations, sharing rain gauges data and carrying out the calibration procedure 1 on the Ku sensors.

Then, we would like to thank the National Weather Service of Ivory Coast - Sodexam - and especially Dr. Aristide Aguia, Dr. Daouda
Konaté and M. Jean-Louis Moulot, for sharing rain gauge data and collaborating with HD Rain on deploying opportunistic rain measurement systems in Abidjan. We thank as well the French DG Trésor, which funded the deployment of HD Rain stations in Ivory Coast through a FASEP project in a partnership with Météo France International, and the Lifi-LED company which installed HD Rain sensors in Abidjan.

We finally thank Drs. Ruben Hallali and Maxime Turko for his advice and proofreading of this paper.





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
