# Peer review of "Enhanced Quantitative Precipitation Estimation (QPE) through the opportunistic use of Ku TV-sat links via a Dual-Channel Procedure"

_Atmospheric Measurement Techniques, 2024_

## Author Comment (AC1)

The authors would like to thank you for your constructive comments, which helped to improve the quality of the paper. Your feedback has been invaluable in refining the manuscript, and we sincerely appreciate the time you took to review it.

PN is time independent in Eq 1 – doesn't it vary in time - with temperature for instance?

**Answer**: Yes, it is true so we add on line 70 : "Additionally, the sensor noise $P_N$ varies with temperature and, therefore, with time. However, these variations are slow compared to the dynamics of the atmosphere. In this study, the sensor noise is considered constant, despite its potential slow variation with the physical temperature of the sensor. The influence of $T_N$ is discussed in Section 4."

**L88-89 :** to be clarified : do you mean that the values of $T_A$ on the one hand and $T_{atm}$ on the other hand do not vary with meteorological conditions ? which central frequencies are assumed to be close to each other ? not clear…..

**Answer**: The sentence "Furthermore, $t_{atm}(t)$ and $T_A(t)$ are assumed to have identical values in various meteorological conditions regardless of the channel used provided that their central frequency is close to each other Barthes et al. (2003)" in replaced by: "Furthermore, in this study we assume that $t_{atm}(t)$ and $T_A(t)$ do note vary significantly over the frequency band used (11 and 12 GHz) and can therefore be considered independent of frequency."

**L96-103 :**

Eq 2 and 3 don't really show the double impact of rain – the dependence of $T_A$ on rain has not been detailed yet …..

**Answer:** Line 96 we replace "Equations (2) and (3) show the impact of rain as highlighted by Giannetti and Reggiannini (2021). Rainfall has a twofold influence:" by "The following sections show how $T_{atm}$ and $T_A$ vary with precipitation, the latter having a twofold influence on the terms of equation 1 (see also Giannetti and Reggiannini (2021) on this subject):"

Line 118, at the end of section 2.1 (Atmospheric transmissivity $t_{atm}$ ) we explain in one sentence how rainfall influences this term : "Finally, rainfall reduces $t_R$, thereby increasing rain attenuation $A_R$ and reducing the total atmospheric transmissivity $t_{atm}$."

Modify last paragraph of 2.2 section: "Scattering effects of rain droplets are considered in simulations (Barthes et al., 2003), where the brightness temperature is studied in relation to atmospheric attenuation for different precipitation rates, showcasing the variability of natural radiation in the atmosphere. Figure 1 shows the *increase* of brightness temperature at 11 and 12 GHz and the corresponding *increase* of atmospheric attenuations for a homogeneous rain layer at different rain rates for a zenith angle of 45°. *Rainfall thus leads to an increase of the antenna radiation temperature.*"

**Eq 11 and 12 :**

I do not understand when and why the dependence of $t_R$ (and tatm) on frequency (as shown in Eq 4, 5 and 6) was dropped

The simplifications made to go from Eq 2 – which as a strong dependence on f through tatm) to Eq 11 which has lost the dependence of tr to frequency have to be explained and justified.

In my understanding Eq 11 should have a term in trA and one in trB

**Answer:**

Indeed, thank you for your comment. We are making a series of changes, detailed below, to justify this simplification in the text.

Correction of equation 6 by replacing:

- $t_R(t)$ → $t_R(f, \theta, t)$
- $t_0(t)$ → $t_0(f, \theta\ t)$
- $A_R(t)$ → $A_R(f, \theta, t)$
- $A_0(t)$ → $A_0(f, \theta, t)$

Modify equation 11 by adding $t_R^A$ et $t_R^B$, $TA_R^A$ et $TA_R^B$ to the numerator

Add to line 160: "Figure 1 shows that in clear sky conditions, for the frequencies considered in the study, the frequency dependencies of $t_0$ and $T_{A0}$ are negligible."

Replace line 161 : "For an ideal LNB with $G_A = G_B$ and $T_{NA} = T_{NB}$ this equation reduces to $t_R$:" by "For an ideal LNB with with $G_A = G_B$ and $T_{NA} = T_{NB}$ and for sufficiently close A and B channel center frequenciers $t_R^A \approx t_R^B$ this equation reduces to $t_R$:"

Modify the paragraph on line 180: "Equation 12 shows that it is necessary to use two channels with different characteristics. Ideally, one should receive a standard satellite **level ($P_E^A > 0$)** while the other one should be tuned to a channel where it receives mainly atmospheric radiation and no (or almost no) satellite signal **($P_E^B \ll P_E^A$)** . This approach helps mitigate any dependency on frequency differences and ensures accurate estimations. This implies as well that the **$t_R$ defined in Eq. (12) corresponds to the $t_R^A$** of Eq. (11) without needing the approximation about tr dependency on frequency."

Add in the paragraph line 315: "As explained previously in Sect.2.3, for the low-cost LNB, there is a gain offset $\Delta G$ between the two channels. When the Ku-sensor does not receive any signal from the satellite ($P_{Sat} = 0$), it is rather simple to estimate this parameter. Under this condition, we can derive from Eq.2 by neglecting the receiver noise."

**L175-179** – The simplification in Eq 11 implies that trchannelA = trchannelB … These paragraph implies that this in not true

I believe some steps are needed for the reader to follow all simplifications leading to simplified Eq 12 from Eq 2 to 6

**Answer:** in previous paragraph if $P_E^B$ is low $t_R^B$ disappears in equation 11. Cf. the answer to the previous remark for details.

**L225-229** : did the authors have a chance to make some comparisons between this ARPEGE iso0° and the one from the MeteoFrance high resolution model AROME ? or with the MeteoFrance radar data (where the iso0° can be extracted using some polarimetric variable)- Same questions with Radiosoundings (or statistics from a satellite based radar such as GPM DPR ) to verify the reliability of those ARPEGE levels ?

**Answer**: This is an idea that needs to be explored. We mention this at the end of the article (cf. line 610 in the original version). We will be working on this in the near future but have not yet done so for this article. In addition, iso0° can be estimated from ground-based or space-based radar observations, but these are not available in real time for the various measurement sites in this study (especially in Ivory Coast). Moreover, as mentioned in the discussion, uncertainty is introduced not only by the height of precipitation, but also by the vertical and horizontal structure of rainfall.

**L229** : it would be nice to have more information about these experiments….
Context/data/results etc…

**Answer**: For simplicity's reason, and to avoid lengthening the paper which already contains experimental results, we feel it is not necessary to introduce further information on this experiment. In any case, this will be the subject of further in-depth work.

**L405 and Fig10**: Please be more precise : what do you lean but correlation between devices ? which data is compared ? time series of rain intensity ? which time step ? any filtering of outliers ?

**Answer**: Modify the legend for figure 10: "Correlations between all the measurements taken Cadarache as a function of the distance between the measuring instruments. The data compared are time series of precipitation rates with a resolution of 30 minutes for 2 and a half months, without correction for outliers.."

Add from line 414: "Given the difference of nature between the rain gauges and the Ku-sensors, obviously, we cannot expect similar values for both instruments. We suppose that these differences are largely mitigated by working at a 30-minutes resolution. On the contrary, it is clear that at high resolution, for instance 5-minutes, we would expect smoother records for the Ku-sensors, which measure spatially integrated over the entire height of the atmosphere." and correction of several elements in the following paragraphs: "Figure 10…" and "Figure 11…".

It would be interesting to see the scatter plots and more statistics on the timeseries (KGE ?) in addition to the QQplots which provide only partial information.

**Answer**: In fact, KGE, NSE, ....many indicators can be used to evaluate or calibrate models precisely, quantifying in a single figure the similarity between observations and simulations of

these observations. Scatter plot allow a point to point comparison. In our case, it's not a question of quantifying the discrepancy between KU-sensor and rain gauge observations, but of assessing whether or not the proposed dual channel retrieval of rain transmissivity applied to Ku-sensor measurement improves the estimation of average rain rate on the link. The problem is that we do not have ground truth for the quantity we are trying to evaluate. Rain gauge measurements are used for statistical comparisons because, in specific cases, precipitation can be observed on the link that is not detected by the rain gauge (e.g., at the start of an event) or vice versa (in the case of very localized events).At best, the inverse algorithm allows for perfect estimation of precipitation over the link; however, these estimations are not strictly identical to those of a rain gauge, which measures very localized precipitation within a few hundred meters. This discrepancy persists even when both types of observations are brought to a 30-minute resolution.

It is not clear from the paper which relationship is used for attenuation-rainfall estimation – are the ITU parameters mentioned in **L191-193** applied to all experiments (dual and single ) ?  Was there any adjustment/calibration ? if yes how ?

**Answer**: Add on line 203 : "For the numerical application, the T-matrix method (Mishchenko, Travis, and Mackowski 1996) is used to calculate the coefficients (alpha and k). Input parameters of this approach include:  the temperature, defined as 10 degrees (mean over the rain column); the drop size distribution model, chosen as the Marshall-Palmer parameterization (Marshall and Palmer 1948); the frequency; the polarization; and the zenith angle between the ground sensor and the satellite."

Change paragraph "Error on k-R relationship": "we use an ITU attenuation - rainfall relationship" replace by "we use marshall-palmer parametrization"

**STYLE/WORDING suggestions**

L3 : link path and not path link : corrected ✓

L7 : which can be commercial (what is meant ? off the shelve ) ? : delated ✓

*THE* measured attenuation : corrected

L19 : what is small-scale or medium-scale rainfall intensity ? are you talking about resolution or intensities values ? Needs to be clarified – If you mean intensities   why single out small and medium , heavy rainfall is the most damaging for 'human/property damage' : ✓ modified "Accurately measuring rainfall intensity is crucial for understanding the water cycle, mitigating human and property damage, and managing water resources"

L21 : Earth Observation satellite (rather than remote sensing ) ? ✓

L35 'previous studies ' – which ones ? delete :  "Previous studies have shown the benefit of using such microwave satellite links for rainfall estimation. A full review of the various problems inherent in this technique can be found in \citet{giannetti2021opportunistic}." ✓

L40 : combination of the signal from the satellite and a background noise that depends on the state… ? ✓ « As a result, the received signal is a mixture of the satellite signal and background noise, which varies with atmospheric conditions. »

L43 : and the baseline signal measured during dry period . In the presence of high rainfall rates, however,… ✓

L47 : 'The present study …. dual channel'. long clumsy sentence – Just get to the point , which is improving rainfall estimation from dual-channel measurement of TV satellite signal by accounting for background noise. No need to repeat low cost etc….✓

L50 : the assumptions it relies on ? (rather than the hypotheses it presupposes) ✓

L52 inherent to rainfall estimation using…✓

L54 : section 2 introduces the physical principles (rather than 'context') ✓

L56 : a physical device whose characteristics …..✓

L60 : physical principles ✓

L61 : ground satellite : you should choose and keep one expression – Earth-Satellite as used before or ground-satellite …..LinkS or a link. ✓

Careful with missing articles along the paper….

L89 : the central frequency of what and compared to what ? not clear – what are each other ?– ✓

check format of the citation.

L104 : Various processes …. Links – do the process characterize the transparency ? do they influence the propagation of the link (or of the microwave signal along the link…) ? To be rephrased. ✓ « Atmospheric transparency is influenced by various processes that affect the propagation of microwave signals along the satellite link. »

L264 : data processing rather than data treatment ✓

L280 : not to go below the threshold (and not exceed …. Which means the opposite) ✓

L314 : Eq 17 and 18 are redundant – you can give directly Eq 18 in dB ….✓

---

## Author Comment (AC2)

The authors would like to thank you for your constructive comments, which helped to improve the quality of the paper. Your feedback has been invaluable in refining the manuscript, and we sincerely appreciate the time you took to review it.

**L23**: "…low revisit time compared to rainfall dynamics." It is somehow clear what is meant here, but it should be formulated more precisely. Please rephrase.

**Answer**: line 21 has been changed: "While satellites can be used to monitor precipitation on a global scale, they require a low Earth orbit to achieve a resolution of a few kilometers, resulting in low revisit time (typically 3-hour average revisit time) compared to rainfall dynamics for which a few tens of minutes are required, especially in convective situations"

**L60**: General comment on Section 2. Each subsection is understandable, but I am missing condensed info on how the improved dual band method is actually applied. It is not directly clear from how the equations are linked in the text, how Delta_G impacts the rain rate estimation via the power law. Maybe there should be an additional subsection that links things together, from t_atm via A to the power law, but explaining how it is done with the Std and the Dual method. Maybe this also fits as an extension of section 2.4.

**Answer**: Following the first review, we have made a few changes to section 2 in order to clarify the assumptions we have made

**Fig 1**: What is the path length (affected by rain-induced attenuation) that is used for the calculation of the attenuation on the y-axis here?

**Fig 1**: Would it be possible to also show the increase in received signal strength for the increased brightness temperature for a given bandwidth, e.g. 1 GHz as used in your LNB?

**Answer**: The received signal strength $P_{atm}$ is linearly related to the brightness temperature (TB), so the curve will look like that of TB but it is in log scale. We change the figure 1 and his caption:

[Figure]

Caption to figure 1: "Sky brightness temperature TB (solid lines), atmospheric induced power $P_{atm}$ at the LNB output (dashed-dotted lines) and atmospheric attenuation (dashed lines) at 11 and 12 GHz for a zenith angle of 45°, with a zero isotherm at 3 km as a function of rain rate for a standard commercial TV-SAT LNB (1 GHz bandwidth, 65 dB gain)"

We have modified lines 145: "Figure 1 shows the variation of the brightness temperature at 11 and 12 GHz as well as the increase of the induced atmospheric signal at the LNB output and the corresponding atmospheric attenuations for a homogeneous rain layer at different precipitation rates for a zenith angle of 45°."

And line 149: "For precipitation rates exceeding about 40 mm/h, a saturation of $P_{atm}$ is clearly observed while the attenuation continues to increase."

**L200**: There is a lot of information provided in the paragraph that starts here, but it is not clear if one of the enhanced models is used, and if not, why?

**Answer:** In addition to taking atmospheric noise into account using the proposed algorithm (which is the central topic), the aim of this paragraph is to highlight the main sources of error that can affect rain rate estimation. For instance, the error in the rain's path length plays an important role (Eq. 15), and we believe it's important to address this issue in a separate paragraph, even if it's not the main topic of the article.

**Section 3 data sets**

**L245**: Since the lower and upper frequency band are directly adjacent, is there power leakage from one band to the other, i.e. for the described case where one TV-satellite only transmits in one of the two bands, how much does still leak into the other band of the receiver where it somehow contaminates the radiometer-like measurement?

**L296**: What does „almost no signal" mean here. How is it different from the setup in France and how does/could it affect the rain rate retrievals?

**Answer**: There are no differences between the two setups, with the exception of the climate in the two regions

**Collective answer**: The potential leakage between the lower and upper frequency bands is quite negligible. First, it's not visible on our spectrum analyzer, suggesting that any leakage is negligible. Secondly, TV channels typically occupy a 30 MHz bandwidth, which makes them highly localized, reducing the likelihood of interference between adjacent channels.

But we think there is another possibility of leakage due to our sensors. The passband filter, designed to isolate the upper or lower frequency bands, may allow a small amount of signal from the band edges to pass through. The sensor would slightly detect signals from the lower band when measuring the upper band.

Or it can be because satellites in adjacent orbital positions can transmit in both bands and provide some signal on the 'radiometric' measurement.

This is why we use the wording 'almost no signal' in the document, which means that the signal in the radiometric band is weak compared to the signal band (at least 10 dB less in the case of clear skies).

**Figure 3**: Is there the potential of leakage of the Astra 19 signals into the receiver of the RS sensors? Or more specifically, what is the half-power beam width, or in general the gain pattern of the antennas? And how high is the noise floor of the radiometer channel of the RS sensors in relation to the potential leakage of Astra 19 signal into the RS receiver via the RS system's antenna side lobes?

**Answer:** The leakage potential between Astra and E5W seems very low, as the half-beam width of the antennas used is close to 1°, whereas the angular distance between Astra and E5W is 35°. Furthermore, the sidelobes are 40 dB below the main lobe, leading to negligible leakage.

**L272**: Is this method with the LSTM documented somewhere in more detail? What is the temporal granularity at which the classification is done?

**Answer:** The method used for the baseline has not been published, as there is a large literature on the subject. As Long short-term memory by Hochreiter and Schmidhuber, 1997, quoted in the paper.

**Figure 3 and Figure 4**: What are the assumed melting layer heights for the plotted path lengths? That would be interesting to know. In Abidjan the elevation angle of the antennas is probably

much higher because of being close to the equators, hence, I expect a much shorter path that is relevant for a typical melting layer height.

**Answer:** As Abidjan is almost below the equator, the elevation angles can range from 0° (eastward or westward links) to about 90° (zenithal links). The 0° isotherm is quite constant around 4.5km. In France the elevation angles are around 40°, with 0° isotherm ranging from 0 (in case of snow) to about 4.5km (in mid-summer).

We add information in the captions: Figure 3: "The satellite link path is identified with colored lines corresponding to the distance between the sensor and the 0° isotherm (here taken at 2000m) in the satellite target " and Figure 4 : "the link path is identified by colored lines corresponding to the distance between the sensor and the 0°C isotherm (here taken to be 4500m) in the satellite target".

And shorten the path link of the Ku sensor targeting SES5 because it was not representative, the angle of elevation is around more than 75°.

**L310 and following**: It is clear from the explanations here and from the shown plots that channel A and B have different P_atm which can be attributed to G_A and G_B. Did you also check that there isn't an offset or some other inaccuracy due to the low-cost electronics of the LNB, which is not optimised to give accurate readings of received signal level?

**Answer:** In figure 5, channel A receives the satellite signal, while channel B only receives atmospheric noise. At around 09:40 (left-hand figure), the antenna was de-pointed for 2 minutes so that both channels received the same atmospheric noise, to evaluate delta_G. The antenna was then returned to its initial position. The existence of an offset is discussed lines 350-355 through a possible difference between $T_N^A$ and $T_N^B$.

We add to line 323: "Then, PB, in Eq. (12), becomes $\hat{P}^B = \alpha P^B$

**Figure 5**: Just a tiny detail, but you could use aligned y-axes here (they are slightly misaligned) and then remove the y-axis tick labels of the plot on the right. Done

**Figure 5**: What is the unit on the y-axis. If it is not dBm, what is the reference level for the dB given here?

**Answer**: Yes, the unit is dBm (same error on figure 6 and figure 7 upper).

**L321**: How do you want to assure that the rain-induced attenuation is strong enough to have t_R approx. 0? Please explain in the text how you identify these events.

**Answer**: We are aware that this procedure is rather empirical and that it is not possible to be sure that $t_R$ is zero. However, in Abidjan there have been extremely heavy rains for which it is easy to identify a plateau showing signal saturation as explained in section 2. Even if $t_R$ is not strictly zero, this guarantees a $t_R$ value close to 0. The procedure is described in lines 370 to 375.

**L339**: Where does the difference of P_A_Tot and P_B_Tot during normal operation (not pointing away form the satellite) come from? Is this due to different transmit power of the satellite in the two bands or can this also be an effect of different gains of the electronics for band A and B?

**Answer:** During normal operations when targeting a satellite (such as at 9:37 in Figure 5), the discrepancy is primarily attributed to variations in the satellite's transmit power (particularly in cases like Figure 5, where the targeted satellite emits minimal signal on one band). However, an additional variation arises from delta_G and Delta_TN, as explained at the conclusion of section 4.1.1.

**L350 and following**: I unterstand the argumentation here on why Delta_G_p3 is used. But doesn't this, the difference of Delta_G depending on what the current brightness temperature is, mean that Delta_G varies with rain rate? If yes, does this affect your results?

**Answer**: Delta_G depends on the LNB's electronics, so there may be slightly different characteristics from one LNB to another. However, as far as we know, there's no reason why delta_G should depend on the rainfall rate.

**L374**: Why does the existence of a dry season „explain the need for sufficient data to calculate Delta_G"? Do you mean that it is harder to get enough data with heavy rain due to the dry season? Please rephrase.

**Answer**: line 381 has been reworded: "In addition, there is a dry season in August and September, which explains the need for sufficient data to calculate the $\Delta G$." by "Furthermore, due to the dry season in August and September with almost no rain, it is necessary to collect data over a period of several months (3 months in this case) to calculate the $\Delta G$"

**Figure 7**: What happened during the period in September 2022 where signal levels for A and B both are increased for several days?

**Answer**: The rise in signal levels (particularly in radiometric mode) is caused by solar radiation. At this inclination and time of year, the sun aligns directly with the sensor.

**Figure 8**: I would put the box plot with Delta_G_ref in the middle. But if you redo this plot, you might consider doing it with something else than boxplots since, here, the spread and distribution of the rainfall sums of the individual Ku sensors is not something we care about, at least not in this plot.

**Answer**: We believe it would be of interest to the reader to examine the impact of delta_G on the variability of rain rate retrievals. For example, Figure 8 shows that the interquartile range is approximately 250 mm for Delta_G - 0.9 and 400 mm for Delta_G + 0.9.

**L388**: This is a bit confusing. Does this mean that the values of Delta_G, as explained in L376 and 377, are used. Or did you do another analysis. Please clarify in the manuscript.

**Answer**: In line 396, we replace: "For the rest of the study, Delta G is estimated by a dedicated analysis for each sensor during saturation events in order to minimize the error on its estimation." by "For the rest of the study, $\Delta G$ is estimated for each sensor using calibration procedure 3 based on a selection of heavy rain events leading to signal saturation."

**L391**: Section 4.2 would maybe benefit from adding two or three subsections when discussing the results since there are different analyses carried out and discussed (gauges vs Ku, Ku SR - Std vs KU SR - Dual).

**Answer**: This section presents many results, but we have chosen not to separate them according to the instruments used. We therefore feel that rewriting it in this way would make it less clear.

**L395**: without correction means that L does not use the +0.360 km (to account for melting layer) and the 0.2 dB for wet antenna? Yes.

Section 2.4 does not specify what "with correction" and "without correction" precisely means, in particular for the melting layer height.

**Answer**: In line 460, we replace "this correction" with "these corrections" for greater clarity.

**L398**: Since you mention that it is important to account for both error sources, wouldn't it be good to show both corrections (melting layer and wet antenna) separately in an updated Figure 9?

**Answer**: In fact, we could have separated these points into two figures. We made this choice because there are already many figures and because these points are not at the heart of the document and will be studied in later research.

**L402**: It is not clear from the figure that the SR estimates are better than the ones from S sensors. In the plot we do not see which rain gauges corresponds to which Ku sensor. Most rain gauges are placed very close to a Ku terminal. Maybe the plot could be optimised to show e.g. each Ku sensor in a separate row of subplots each only with the rain gauges in the vicinity of its location or its path.

**Answer**: The main purpose of this figure is to show that corrections play an important role in some situations and that it is important to take them into account. Moreover, the dual-channel procedure is more interesting for heavy rain; it does not improve the estimation much in the presence of not very heavy rain as is the case in this figure

**L406**: What does HDR mean here? Probably HD Rain. But this abbreviation was not introduced.

**Answer**: We made a mistake in the wording, HDR station means Ku sensors: "It can be seen that the correlation between the *Ku sensors* is mainly above the trend line (84%), indicating consistency between *the sensors."*

**L408**: One reason why the correlation between rain gauges might drop faster with increasing distance compared to the Ku sensors is that the Ku sensors provide a path-averaged rain rate estimate which smoothens spatial extremes compared to the rain gauge measurements. This should be mentioned here in the text, because now the text sounds as if the gauges are inferior devices for rainfall measurement with the statement in the sentence before about the consistency of the HDR devices.

**Answer**: The sentence line 420: In contrast, the correlation for the rain gauges, which provide a direct measurement, is fairly heterogeneous, falling below 0.5 at a difference of 4 km

Is replaced by:

On the other hand, and as expected given the point measurements of the rain gauges, the correlation is fairly heterogeneous, falling below 0.5 for a distance of 4 km.

**L415-L417**: I do not unterstand the argumentation in these two sentences. Please rewrite.

**Answer**: The sentence: "Given the difference of nature between the rain gauges and the Ku-sensors, obviously, we cannot expect similar values for both instruments. We suppose that these differences are largely mitigated by working at a 30-minutes resolution. On the contrary, it is clear that at high resolution, for instance 5-minutes, we would expect smoother records for the Ku-sensors."

Is replaced by:

"Given the difference in spatial resolution between rain gauges and Ku sensors, we cannot expect similar values for both instruments. We assume that this difference in spatial resolution can be mitigated by integrating over time the precipitation rates measured by each device."

**Figure 11**: Is this done with data from all gauges and all Ku sensors (separated by the applied method) or done with one pair of gauge and Ku sensor?

**Answer**: The data includes all gauges and all Ku sensors.

**L425**: What is a "directing coefficient"? Please clarify in the manuscript what is calculated here.

**Answer**: The sentence: We compute the linear regression line and find a directing coefficient of 0.96, which is very close to the trendline (ideal curve in black dotted lines).

Is replaced by:

The slope of the linear regression is 0.96, which is very close to the trendline (ideal curve in black dotted lines)

**L428**: I guess you mean "S-Std" here and not "SR-Std" based on what is described here. If not, I understood things wrongly. But maybe the text could be clearer.

**Answer**: The sentence is correct, but some things need to be clarified: SR-Std means that an SR Ku device is used but with the std algorithm instead of the dual algorithm.

The caption of figure 11 is modified: "Quantiles (mm/h) from the 1st to the 100th percentile (colored points) of 30 min resolution records for Ku-sensors using dual algorithm (red) and std algorithm (green and light red) vs. rain gauges (after excluding days when none of the devices detect rainfall)."

**L455**: Why are the satellite signals received in Ivory Coast much weaker? Please explain in the text.

**Answer**: In Ivory Coast, from the satellites are weaker than main satellites used in Europe (Astra 19 and Hotbird) because their EIRP (Equivalent Isotropically Radiated Power) is lower

**L484 (and following sentences)**: "…as the rain may be too light to be detected by the rain gauge". Since the satellite link rainfall estimation also has a lower detection limit I would not agree with this argumentation. If you want to use this argument, please provide info on the lower threshold of the rain gauge data and of your rainfall estimates. A more likely cause for these false-positive rainy days could be that the rain event detection method, briefly described in section 3.2 but not explicitly validated, might produce false-positive rain events. This is a common challenge when processing attenuation data from terrestrial microwave links for which the raw data time series look very similar to the ones from satellite microwave links. Please elaborate on this and/or updated the text.

**Answer**: Yes, we agree with your remark. Another phenomenon leading to such cases is the heterogeneity of the rain. We have therefore replaced line 495: "This is probably due both to the heterogeneity of the rain (rain passing somewhere above the link but not at the location of the rain gauge) and to one-off errors in the rain detection algorithm leading to false positives."

**Figure 13**: These plots should be larger. ok

L552: "…given their difference of nature" is not very precise. You probably mean the different spatial integration characteristics and different operating principle in general. But it should be more precise in the text. Also, why exactly do we expect that the lowest quantiles are overestimated by the Ku-sensors?

**Answer**: We add to line 565: 'Ku-sensors records the mean rainfall over a few kilometers long link while rain gages do punctual measurements. This will lead to smoother records for Kusensors, with more rain occurrence and so larger small-quantiles and less heavy rainfall and so smaller high-quantiles'

**L556**: "…but also that the intensity-dependence of this underestimation seems to have been solved". Please be more precise in the text. I do not understand what is meant here.

**Answer**: We modify line 571 with "However, it also demonstrates that while the sensor error initially showed a strong dependence on rainfall intensity (as seen in the increasing green curve in Fig. 16), this dependency is significantly reduced after correction, with the blue curve remaining relatively stable between 1.5 and 2."

L559: You could cite Polz et al. (2023), which you already cited in the introduction, again here because they have analysed this effect in detail for terrestrial microwave links.

**Answer**: We did on line 577.

L581: This is the first time I read about "quasi vertically pointing" in the manuscript. This should be either explained here, or maybe better, in the section describing the setup in Abidjan.

**Answer**: We have added the elevation angles in sections 3.3 and 3.4

**Figure A1**: If I understand the caption correctly, I would name the data shown in the plot on the top "FP" for false-positive (gauge has no rain, Ku detectors rain) and the data shown in the plot at the bottom "FN" false-negative (gauge has rain, Ku detects no rain).

**Answer**: we have modified the figure A1 and its caption.

**Technical corrections (only partly documented, mainly done for section 4 and 5, due to limited time spent on this task):**

L3: "link path" instead of "path link" ✓

Equation 2: Should appear at end of sentence and not a top of the page.

L192: I have not seen the word "lineic" been used a lot in this context. You might consider writing "path attenuation". In the case of equation 13 here it is the "specific path attenuation". ✓

L413: The figure caption says that 30-minute resolution data is used. Here you write 1h. Please correct. ✓

L429: Delete the „of in „of the atmospheric… " ✓

L443: Better write „analysis" instead of „study" here because you only refer to the results of this section and not the results of the whole manuscript. ✓

L448: Write „satellite with low signal strength". ✓

L454: There is a „was" or „is" missing in this sentence. ✓

L512: Unclear what „by both of the rain gauges…" means here. Please rephrase. ✓

L527: I do not see „red crosses" in Figure 14. I guess it should be „black crosses" here. ✓

L528: Same here. No „cyan dots", maybe should be „blue dots". ✓

L545: Write „…more homogeneous rainfall distribution along the path affected by rain" or something similar. ✓

L547: From the text it seems Fig 15 and Fig 16 are not the ones that should be refered to here. ✓

Fig 15: y-axis should maybe not be called „Station quantiles" but something like „Ku-sensor quantiles". On the x-axis write „gauge" instead of „gage". ✓

Fig 16: „HDR" on the y-axis is not used in the text except for two individual occasions. Maybe use something else here. ✓

L573: write „rain rate" instead of just „rain". ✓

L600: you maybe meant „because of the power-law" instead of „but power-law…" ✓

---

## Author Response (AR2)

Collective response

**Notes on figure 1:**

- Fig 1: It is good to see P_atm in this plot now. One minor, but important, improvement would be to use the same scaling for A_r and P_atm on the two y-axis, because these are the two „competing" processes and it would be easier to judge which process is the dominating one at which rain rates. P_atm could also be scaled so that it starts at the bottom of the plot for zero rain rates.
- the authors should add in Fig 1's caption and in thetext how Fig 1 was obtained (from simulations I guess - but it should be clarified)

**Answer :** done

[Figure]

Caption: *Simulated* sky brightness temperature TB (solid lines), atmospheric induced power Patm at the LNB output (dashed-dotted lines) and atmospheric attenuation (dashed lines) at 11 and 12 GHz for a zenith angle of 45°, with a zero isotherm at 3 km as a function of rain rate for a standard commercial TV-SAT LNB (1 GHz bandwidth, 65 dB gain).

**Fig 3 and 4**: Why are the paths only plotted for one site in Fig 4? If this is to make the plot less busy, that is okay. But it should be mentioned in the caption that paths are only shown for one site. Alternatively, you could, of course, plot all paths as it is done in Fig 3.

**Answer**: the link path is identified by colored lines corresponding to the distance between the sensor and the 0°C isotherm (here taken at 4500m) in the satellite target *(for legibility reasons, only 2 paths have been drawn as examples)*

From Fig 5 and given the Y-axis ticks, deltaG seems to be at least 2.5 dB (> one graduation) while the values 1.85dB and 1.87dB are provided in the text.
The authors should verify ...

**Answer**: After verification, the values are 1.87dB and 1.89dB and have been calculated on the figure.

Regarding your response to my comment on „L350 and following "about the difference between Delta_G_p1 and Delta_G_p3: You write in your response that „there's no reason why delta_G should depend on the rainfall rate". But your results for Delta_G_p1 (obtain during clear sky) and Delta_G_p3 (obtained during heavy rain) show that there can be a difference of 0.9 dB for Delta_G, which might be relevant in the rainfall retrieval process. This could be caused by differences of the reported signal level (compared to the correct signal level) of the LNB for different magnitudes of received signal level, i.e. the LNB might report signal power at -32.0 dBm while the real received signal power is -32.5 dBm and it could report -28.0 dBm while the real value is -28.9 dBm (that might be a bit exaggerated). This effect could also be different for different frequencies. This could be checked in the lab. My gut feeling is that the used electronics should not be that bad. But, in the absence of any other explanation for the shown differences between Delta_G_p1 and Delta_G_p3 (or maybe I just missed it in the manuscript), this could be considered. Note that I do not suggest that the authors look into this now for the revision of the manuscript. My comment is just meant as I suggestion for an explanation of the observed differences in the Delta_G value

This paragraph explains the difference between the two procedure: "The difference between both procedures varies from 0.9dB to 0.2dB. Equation 12 assumes that $T_N^A$ and $T_N^B$ are equal and that the radiation produced by the sensor is negligible. Except that this assumption is wrong, and after a few experiments we estimate the difference between $T_N^A$ and $T_N^B$ (= Delta $T_N$) to be < 15K. Furthermore, it can be seen that when the brightness temperature $T_A$ is very low, as in procedure 1, the channel-dependent brightness temperatures of the noise are no longer negligible. Whereas in procedures 2 and 3, $T_A$ is high (>170 K) so the difference Delta $T_N$ is negligible. We therefore use the value from procedure 3."

Fig 9: The y-axis should be equal on both subplots: Done

[Figure]

[Figure]